# MetaFormer with Holistic Attention Modelling Improves Few-Shot Classification

## Abstract

Pre-trained vision transformers have revolutionized few-shot image classification, and it has been recently demonstrated that the previous common practice of meta-learning in synergy with these pre-trained transformers still holds significance and contributes to further advancing their performance. Unfortunately, the majority of working insights in meta-learning such as task conditioning are specifically tailored for convolutional neural networks, thus failing to translate effectively to vision transformers. This work sets out to bridge this gap via a coherent and lightweight framework called MetaFormer, which maintains compatibility with off-the-shelf pre-trained vision transformers. The proposed MetaFormer consists of two attention modules, i.e., the Sample-level Attention Module (SAM) and the Task-level Attention Module (TAM). SAM works in conjunction with the patch-level attention in Transformers to enforce consistency in the attended features across samples within a task, while TAM regularizes learning of the current task with an attended task in the pool. Empirical results on four few-shot learning benchmarks, i.e., miniImageNet, tieredImageNet, CIFAR-FS, and FC100, showcase that our approach achieves the new state-of-the-art at a very modest increase in computational overhead. Furthermore, our approach excels in cross-domain task generalization scenarios.

## 1 Introduction

There has been a sustained focus on few-shot learning (Vinyals et al., 2016b; Snell et al., 2017) where only a few labeled samples (support) are given for predicting unlabelled samples (query), aiming to approach human-level intelligence that can rapidly grasp new concepts. Meta-learning (Thrun & Pratt, 2012) has been a de-facto approach in dealing with few-shot learning, via leveraging the knowledge learned from previous tasks (Finn et al., 2017; Raghu et al., 2019). Recently, pre-trained Vision Transformers (ViTs) have impressively rivaled traditional Convolutional Neural Networks (CNNs) across diverse vision tasks (Dosovitskiy et al., 2020; Liu et al., 2021; Zhu et al., 2020; Ranftl et al., 2021; Strudel et al., 2021). Their impact has extended to few-shot image classification as well (He et al., 2022b; Dong et al., 2022; Lin et al., 2023). More notably, recent research suggests that meta-learning can effectively synergize with these pre-trained transformers to further enhance their few-shot learning performance (Hiller et al., 2022; Hu et al., 2022).

Despite the initial success by directly adapting ProtoMAML (Triantafillou et al., 2020) and Proto-Net (Snell et al., 2017) in Hiller et al. (2022); Hu et al. (2022), the potential of leveraging other essential meta-learning advancements such as conditional meta-learning (Yao et al., 2019; Garnelo et al., 2018) that accommodates a diverse range of tasks remains unexplored in the context of ViTs. The core idea behind conditional meta-learning is to learn the relationship between tasks through task embeddings (Yao et al., 2020; Zhou et al., 2021a; Jiang et al., 2022), so that the transferable knowledge shared among only closely related tasks improves generalization. Accurately modeling task relationships under ViTs, however, poses a non-trivial challenge due to the expensive computation costs involved. For instance, considering $n$ patches in an image and $NK$ support images in a $N$-way $K$-shot task, the holistic attention across a total of $N^T$ tasks could have a time complexity of up to $\mathcal{O}((nNKN^T)^2)$. Given the oftentimes huge $N^T$ as the number of episodic tasks sampled, the straightforward holistic attention becomes prohibitively expensive.

In this work, we are motivated to propose a novel ViT-backed framework dubbed MetaFormer, which plays the strength of self-attention tailored for meta-learning while avoiding substantial computation overhead. Specifically, we break down the holistic attention into two stages, i.e., intra-task and inter-task interactions. In the first stage, we propose the Sample-level Attention Module (SAM) to accurately and efficiently model sample relationship within a task. By separately applying spatial attention implemented by the original ViT modules and sample attention in multiple layers, SAM alleviates high computational complexity and captures coarse-to-grain sample relationship. We implement the sample attention by a sample-wise attention mask, which not only enhances the consistency in identifying task-specific discriminative features but also facilitates the extension to autoregressive inference that takes interactions between query samples into consideration. Secondly, we propose the Task-level Attention Module (TAM) to model inter-task interactions. TAM automatically learns a task-specific probe vector, which summarizes the discriminative patterns of a task. Based on the probe vector, TAM retrieves the most relevant semantic feature patterns from seen tasks to regularize learning of the current task. To combat against the huge number of historical tasks, TAM consolidates probe vectors of previous tasks into a dynamic knowledge pool for retrieval. By stacking the SAM and TAM with the original ViT modules to formulate holistic attention, MetaFormer fully exploits knowledge within and across tasks and thereby demonstrates significant performance gains on established few-shot learning benchmarks.

The main contributions of our work are summarized as follows:

- We propose MetaFormer, a ViT-backed meta-learning method that takes full advantage of transformer characteristics for few-shot image classification and remains compatible with state-of-the-art pre-trained ViT backbones.
- We introduce an autoregressive few-shot image classification setting to leverage query relationships and show our method can be easily extended to this setting via the sample attention mask.
- Extensive experiments demonstrate that our MetaFormer outperforms state-of-the-art meta-learning methods on four widely-used few-shot learning benchmarks, including miniImageNet (Vinyals et al., 2016b), tieredImageNet (Ren et al., 2018b), CIFAR-FS (Bertinetto et al., 2019), and FC100 (Oreshkin et al., 2018). Also, it achieves remarkable performance in eight cross-domain benchmarks (Oh et al., 2022) and multi-domain benchmarks (Triantafillou et al., 2020).

## 2 RELATED WORK

**Meta-Learning in Few-Shot Classification.** Meta-learning serves as a fundamental framework for few-shot learning with the aim of transferring prior knowledge for quickly adapting to new unseen tasks. Most related to our work are metric-based and generation-based meta-learning methods. Metric-based methods (Vinyals et al., 2016b; Snell et al., 2017; Oreshkin et al., 2018; Lee et al., 2019; Chen et al., 2021a; Zhang et al., 2020a; Ma et al., 2021c; Simon et al., 2020) seek to embed samples into global universal feature representations and use some nearest neighbor algorithm to measure sample similarity in the embedding space to give predictions. However, fixed embedding is not very robust and sufficient to accommodate tasks with significant shifts due to cluttered backgrounds and intricate scenes. To adapt the feature embedding to new tasks, several approaches are proposed to perform task adaptation utilizing within-support (Rusu et al., 2018; Ye et al., 2020) and support-query (Xu et al., 2020; Hou et al., 2019; Doersch et al., 2020; Kang et al., 2021) sample relationship. Besides, parameter-generation methods directly generate task-conditioned parameters for task adaptation (Qiao et al., 2018; Ma et al., 2021b; Sun et al., 2021; Bertinetto et al., 2016; Gidaris & Komodakis, 2019; Munkhdalai et al., 2018; Cai et al., 2018), such as convolution kernel parameters (Ma et al., 2021b; Zhmoginov et al., 2022) and batch normalization parameters (Requeima et al., 2019; Bateni et al., 2020). However, many methods such as task conditioning are specially tailored for CNNs and thus fail to translate effectively to vision transformers. Our approach is dedicated to fully leveraging the attention characteristics of vision transformers for intra- and inter-task interactions.

**Inter-task knowledge sharing.** Meta-learning objective is to organize and distill previous knowledge for future reuse when adapting to new unseen tasks. To handle tasks with different distributions, a handful of works built upon the gradient-based methods try to extract the underlying task structure for customizing initialization (Yao et al., 2019; 2020; Zhou et al., 2021a; Jiang et al., 2022). However, these algorithms rely on time-consuming clustering and the discriminative task representations are difficult to learn (Jiang et al., 2022). In this paper, we adopt the knowledge pool to learn structured

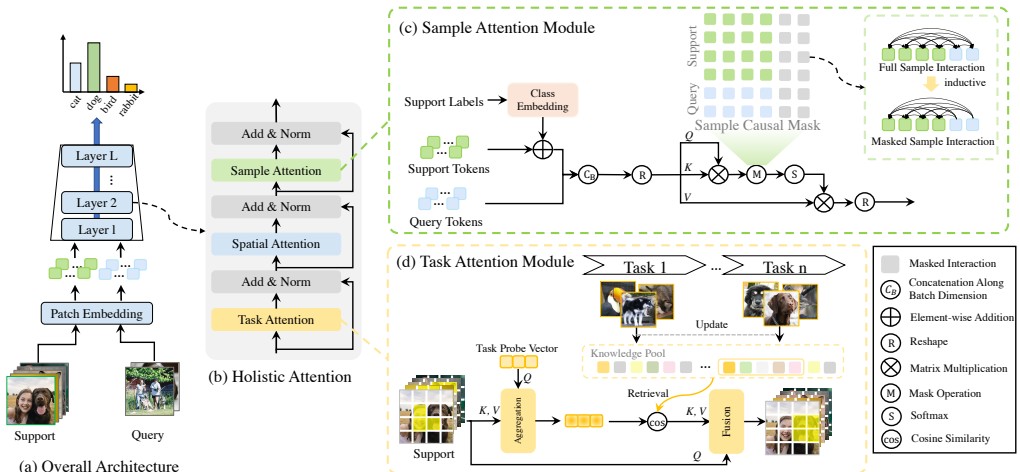

Figure 1: **Overview –** (a)The architecture of the MetaFormer with holistic task attention built upon the original Vit Block, Sample Attention Module (SAM), and Task Attention Module (TAM), which extracts feature representations of support and query samples with only once feedforward while following the inductive protocol; (b) Three modules build holistic attention integrated with intra- and inter-task interaction in a sequential mode; (c) Schematic illustration of the proposed Sample-level Attention Module (SAM) with sample causal attention mask to exploit the within-support and support-query relations for task-specific embeddings. (d) Schematic illustration of the proposed Task-level Attention Module (TAM) where foreground region is further concentrated by previous relevant semantic knowledge with the task-specific probe vector.

meta-knowledge (i.e., key feature patterns), which is then tailored to the current task through attention-based aggregation. We show later this achieves a better trade-off between task-specific and task-agnostic knowledge sharing. Recent works (Wang et al., 2022a; Smith et al., 2023; Douillard et al., 2022; Wang et al., 2022b) propose the prompt pool and use inter-task attention for continual learning settings to prevent catastrophic forgetting, which is different from our meta-learning setting.

**Vision Transformers in Few-Shot Learning.** Vision Transformers (Dosovitskiy et al., 2020; Liu et al., 2021; Tu et al., 2022) utilize the self-attention mechanism to encode long-range dependency in the data. There's a growing inclination in recent works towards designing self-distillation pretraining to train few-shot Transformers (He et al., 2022b; Dong et al., 2022; Lin et al., 2023). For example, HcT (He et al., 2022b) utilize the DINO-based (Caron et al., 2021) teacher-student framework to distill the global class token and train three cascaded transformers with two pooling layers in between. To further supervise the patch tokens, SUN (Dong et al., 2022) adopts the patch-level pseudo labels generated by the teacher network and SMKD (Lin et al., 2023) introduces the patch reconstruction loss in Masked Image Modeling (MIM) (He et al., 2022a; Bao et al., 2022; Zhou et al., 2022a). These methods seek a generalizable feature embedding that is fixed for different tasks. However, previous meta-learning methods (Hu et al., 2022; Hiller et al., 2022) have shown that meta-learning is beneficial for transferring past knowledge for feature adaptation. FewTURE (Hiller et al., 2022) learns the support-aware patch importance mask in the inner loop to mitigate the supervision collapse issue. Yet they use this in the top classifier, giving the network no or less opportunity to refine the features to adapt to a new task with a large variance. Thus there is an opportunity to further develop the meta-learning framework specifically for playing the strength of vision transformer. Similarly, we also ground our proposed meta-learning method on pre-trained vision transformers but embed the sample relationship into ViT with hierarchical task attention to learn more discriminative features for each task. And thus our contribution is orthogonal to SKMD.We empirically show that MetaFormer can further improve the joint performance.

## 3 METAFORMER FOR FEW-SHOT CLASSIFICAITON

We present our approach in this section. The overall architecture of our MetaFormer is illustrated in Figure 1. We start by briefly introducing the few-shot image classification setting and the self-

attention of vision transformer in Section 3.1 and then elaborate on our proposed Sample-level Attention Module (SAM) and Task-level Attention Module (TAM) in Section 3.2 and Section 3.2, respectively. Finally, using SAM and TAM as the core building block, we present a new vision transformer with holistic attention for meta-learning in Section 3.4.

## 3.1 PRELIMINARIES

**Problem formulation.** Few-shot learning aims to learn a model that can adapt to recognize new classes with only a few labeled examples. We adopt the episodic training manner following previous works (Vinyals et al., 2016b; Hiller et al., 2022). In a classical $N$-way $K$-shot setting, each episode randomly selects $N$ classes to form the support set $\mathcal{C} = \{(x_i^c, y_i^c)\}_{i=1}^{N \times K}$ containing $K$ samples in each class and the query set $\mathcal{T} = \{(x_j^t, y_j^t)\}_{j=1}^{M}$ with $M$ samples. Predictions are independent for every query sample for the inductive protocol (Vinyals et al., 2016b). We also introduce the autoregressive setting from regression tasks (Nguyen & Grover, 2022; Bruinsma et al., 2023) to classification tasks, where we autoregressively predict query samples and allow interactions between subsequent query samples and those predicted earlier.

**Self-attention in Vision Transformers.** Given a $N$-way $K$-shot task as input $\mathbf{X} \in \mathbb{R}^{(NK+M) \times H \times W \times 3}$, ViTs (Dosovitskiy et al., 2020) first divide individual images into $n$ non-overlapping patches and then map them into $d$-dimension tokens through a linear projection layer. After that, a trainable class token is prepended as the final input token sequence $\mathbf{X} \in \mathbb{R}^{(NK+M) \times L \times d}$ ($L = n + 1$), taken by several multi-head self-attention (MSA) layers and MLP layers for feature extraction. Consider a MSA layer with $H$ heads, and query, key, and value embeddings of the input $\mathbf{X}$ are given as $\mathbf{Q} = W^Q \mathbf{X}$, $\mathbf{K} = W^K \mathbf{X}$, $\mathbf{V} = W^V \mathbf{X}$, respectively. The output of MSA is given as:

$$\text{MSA}(\mathbf{Q}, \mathbf{K}, \mathbf{V}) = \text{Concat}(h_1, \ldots, h_H) W^O$$
$$\text{where } h_i = \sigma(\mathbf{A}_i) \mathbf{V}_i = \sigma\left(\frac{\mathbf{Q}_i \mathbf{K}_i^\top}{\sqrt{d_k}}\right) \mathbf{V}_i \quad (1)$$

where $W^O$ is the output projection matrix; $d_k = d/H$ is the head dimension; $\sigma(\cdot)$ denotes the softmax activation function; $\mathbf{A}_h \in \mathbb{R}^{L \times L}$ is the attention matrix measuring pairwise token affinity at different spatial locations. After MSA as equation 1, every token within the image is aware of the global spatial information and thus we term MSA as the spatial attention module.

## 3.2 SAMPLE-LEVEL ATTENTION MODULE

To facilitate sample correspondence learning for task adaptation, most existing methods usually incorporate extra modules on top of the feature extractor with global feature embedding (Ye et al., 2020; Doersch et al., 2020; Hiller et al., 2022). However, it is demonstrated that different layers of the backbone yield different semantic levels of feature embedding and thus different types of knowledge (Raghu et al., 2021). Motivated by this, we propose to leverage coarse-to-grain multi-scale information across layers to capture discriminative sample interactions at the patch token level.

**Joint Space-Sample Attention.** A straightforward and intuitive approach is to perform self-attention over both the spatial and sample dimensions simultaneously. Given the task input $\mathbf{X}$, the core computation of one MSA layer primarily revolves around calculating the attention matrix $\mathbf{A}_J \in \mathbb{R}^{(NK+M)L \times (NK+M)L}$ in equation 1. Therefore, the complexity of the joint space-sample attention is $O((NK+M)^2 L^2)$. Such joint space-sample interaction empowers the vision transformer to capture sample relationships for task-specific embedding, but it comes at a high computational cost and incurs heavy memory footprints.

**Decoupled Space-Sample Attention.** To alleviate the computational complexity, we propose a more efficient architecture designed to decouple spatial attention and sample attention, illustrated in Figure 1(b). In the case of decoupled space-sample attention, within each layer, our approach initially computes spatial-only attention as equation 1 to obtain features isolating backgrounds and emphasizing underlying objects. Subsequently, we reshape the token sequence to $\mathbb{R}^{L \times (NK+M) \times d}$ that is fed to MSA with sample attention matrix $\mathbf{A}_S \in \mathbb{R}^{(NK+M) \times (NK+M)}$, incorporating sample interactions across all patches at the same spatial location to capture the similarities and variances among samples, which is essential for the feature extraction in a given task to discern task-specific

discriminative regions. As such, the computation complexity is reduced to $O(L(NK + M)^2 + (NK + M)L^2)$. See Figure 5 in Appendix B for an illustration. Though this decoupling shares the spirit with video transformers (Ho et al., 2019; Bertasius et al., 2021), it is crucial to highlight that our consideration of the sample-to-sample relationship in few-shot learning presents a unique challenge distinct from the frame-to-frame relationship in videos, i.e., query samples have to be differentiated from support ones. This challenge motivates the following introduction of sample causal masks.

**Sample-level Attention Module (SAM).** As shown in Figure 1(c), we introduce our Sample-level Attention Module (SAM) with label infusion and the designed causal masking mechanism to further enforce consistency in the attended features across samples within a task. We first get the embedded support category information $W^c y \in \mathbb{R}^{1 \times d}$ via the linear projection matrix $W^c$, which is infused to support tokens through the elementwise addition. For the obtained sample attention $\mathbf{A}_S$, we maintain the sample causal mask $\mathbf{H} \in \mathbb{R}^{(NK+M) \times (NK+M)}$ to restrict the sample interaction patterns as:

$$\hat{\mathbf{A}}_S = \mathbf{A}_S \odot \mathbf{H} \tag{2}$$

where $\odot$ is the element-wise product.

Through the constraint, support samples can attend themselves to strengthen intra- and inter-class discriminative clues, which query samples utilize for task-specific feature consistency learning. Note that this mask mechanism also makes our method comply with the inductive protocols. In the autoregressive scenario, we also extend our SAM with the autoregressive causally-masked sample attention to embed the query-query interactions into the vision transformer. Figure 2(b) shows an example mask with $N = 4$ and $K = 1$. Query samples attend to support samples and earlier predicted queries in an autoregressive fashion, which thus serves to implicitly expand the support set for subsequent query predictions.

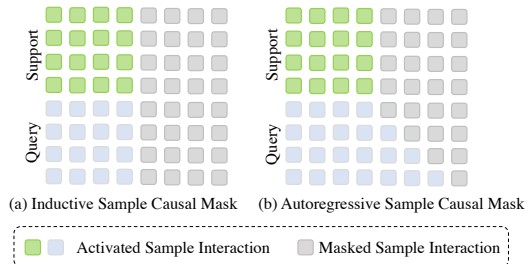

(a) Inductive Sample Causal Mask    (b) Autoregressive Sample Causal Mask

☐ ☐ Activated Sample Interaction    ☐ Masked Sample Interaction

Figure 2: Sample Causal Attenntion Mask. An example with $N = 4$ and $K = 1$. (a) Inductive sample causal mask for within-support and support-query sample correspondence learning. (b) Autoregressive sample causal mask for extra query-query sample correspondence learning.

### 3.3 TASK-LEVEL ATTENTION MODULE

In this section, we introduce the details of the proposed Task-level Attention Module (TAM), as illustrated in Figure 1(d). The goal of TAM is to transfer previous task knowledge for regularizing the adaptation in the current task. To this end, we introduce a knowledge pool consolidated during meta-training to organize learned knowledge. When a new task comes, we first acquire the task-specific probe vector to represent the current task. It taps into the knowledge pool to retrieve relevant knowledge from historical tasks, which is fused to enhance the support feature representations. We elaborate on four key components as follows: task probe vector aggregation, knowledge retrieval, pool consolidation, and knowledge fusion.

**Task Probe Vector Aggregation.** Given a task consisting of support and query sets, we first gather the task information with learnable task probe vectors $\mathbf{G} \in \mathbb{R}^{T \times d}$, which are computed along with support patch tokens $\mathbf{X}_\mathcal{C}$ to aggregate the key parts of samples and the whole task representations. Specifically, we perform the task aggregation using attention as:

$$\text{Aggregation}(\mathbf{Q_G}, \mathbf{K_{X_c}}, \mathbf{V_{X_c}}) = \text{MSA}(\mathbf{Q_G}, \mathbf{K_{X_c}}, \mathbf{V_{X_c}}) \tag{3}$$

where $\mathbf{Q_G}$ is query embedding of task probe vectors; $\mathbf{K_{X_c}}$ and $\mathbf{V_{X_c}}$ are key and value embeddings of support patch tokens, respectively. This allows task probe vectors to focus on relevant task-specific feature patterns and ignore irrelevant semantics of each sample.

**Knowledge Retrieval.** After gathering the task information, we retrieve relevant knowledge using a simple weighted summation strategy from the knowledge pool $\mathbf{P} \in \mathbb{R}^{Z \times d}$ with $Z$ components (which will be introduced below). The retrieval is formulated as:

$$\mathbf{R} = \sum_z \gamma(\mathbf{G}, \mathbf{P_z}) \mathbf{P_z} \tag{4}$$

where $\gamma$ is the score function based on cosine similarity between task probe vectors and pool components. $\mathbf{R} \in \mathbb{R}^{T \times d}$ is the retrieved historical knowledge that can be thought of as key feature semantics (e.g., ears and eyes of the dog) related to the current task samples.

**Pool Consolidation.** During meta-training, we maintain a knowledge pool $\mathbf{P}$ updated by every sequentially coming task. To consolidate the learned knowledge in the pool, we select relevant components from the pool and integrate them with new information brought by the current task as follows:

$$\mathbf{P}_s = \mathbf{P}_s + \mathbf{G}_i \tag{5}$$

where $s = \arg\max \gamma (\mathbf{G}_i, \mathbf{P})$ representing components most similar to $i$-th task probe vector. This method also allows us to control the pool size and the memory consumption.

**Knowledge Fusion.** To regularize the adaptation of new tasks with historical knowledge, we deliver the union of original task vectors $\mathbf{G}$ and retrieved knowledge $\mathbf{R}$ to enhance the support patch token representations via the attention mechanism as follows:

$$\text{Fusion}\left(\mathbf{Q}_{\mathbf{X}_\mathcal{C}}, \mathbf{K}_{[\mathbf{G};\mathbf{R}]}, \mathbf{V}_{[\mathbf{G};\mathbf{R}]}\right) = \text{MSA}\left(\mathbf{Q}_{\mathbf{X}_\mathcal{C}}, \mathbf{K}_{[\mathbf{G};\mathbf{R}]}, \mathbf{V}_{[\mathbf{G};\mathbf{R}]}\right) \tag{6}$$

where $\mathbf{Q}_{\mathbf{X}_\mathcal{C}}$ is query embedding of support patch tokens and $\mathbf{K}_{[\mathbf{G};\mathbf{R}]}$ and $\mathbf{V}_{[\mathbf{G};\mathbf{R}]}$ are key and value embeddings of regularized task-specific semantics, respectively. The intuition here is to leverage well-learned feature semantics in previous similar tasks to strengthen discriminative regions in the new task.

## 3.4 METAFORMER WITH HOLISTIC ATTENTION

Using SAM and TAM as the basic building blocks working in conjunction with original ViT modules, we propose a new vision transformer with holistic attention, named MetaFormer $f_\theta$, customized for meta-learning in the few-shot image classification. Holistic attention incorporates both the intra- and inter-task interactions at different semantic levels to extract rich task-specific feature representation and thus adapt to new tasks more effectively. Built upon feature embedding extracted by MetaFormer, we estimate the class patch prototypes by averaging support patch tokens per class $p_k = \frac{1}{|\mathcal{C}^k|} \sum_{x \in \mathcal{C}^k} f_\theta(x)$. Query samples are predicted based on patch-wise cosine similarity with prototypes (Lai et al., 2022; Hiller et al., 2022). The probability of $k^{th}$ category is:

$$P(\hat{y}_t = k \mid x_t) = \frac{e^{d(f_\theta(x_t), p_k)/\tau}}{\sum_c e^{d(f_\theta(x_t), p_c)/\tau}} \tag{7}$$

where $d$ indicates the cosine distance and $\tau$ is scaling temperature. The cross-entropy loss function with the few-shot label $y_t$ is:

$$\mathcal{L}_{\text{CE}} = -\sum_{t=1}^{M} \log P(\hat{y}_t = y_t \mid x_t) \tag{8}$$

**Autoregressive Inference.** In the autoregressive setting, we propose to enrich the support prototypes by feeding previously predicted queries as the auxiliary support set $\mathcal{Q}$ with predicted probability belonging to class $k$. We take $P(\hat{y}_t = k \mid x_t)$ as sample weights and estimate auxiliary prototypes in a weighted average manner $\hat{p}_k = \frac{1}{\sum_{x \in \mathcal{Q}^k} P(k|x)} \sum_{x \in \mathcal{Q}^k} P(k \mid x) f_\theta(x)$. Then new prototypes can be updated by the mean of $p_k$ and $\hat{p}_k$. Furthermore, considering modeling dependencies between all $M$ query samples requires $M$ prototype updates. Alternatively, we introduce $r$ sampling size of queries at a time to achieve faster and more consistent prototype updates.

## 4 EXPERIMENTS

We evaluate the effectiveness of our proposed MetaFormer on few-shot image classification tasks, including standard in-domain few-shot learning in Section 4.1 and conduct a broader study of cross-domain and multi-domain few-shot learning in Section 4.2. Additionally, we conduct ablation studies to verify the effectiveness of the proposed holistic attention modeling with Sample-level Attention Module (SAM) and Task-level Attention Module (TAM) in Section 4.3 and Section 4.4.

## 4.1 STANDARD FEW-SHOT LEARNING

**Datasets.** We train and evaluate our MetaFormer on the four standard few-shot benchmarks: *mini*ImageNet (Vinyals et al., 2016b), *tiered*ImageNet (Ren et al., 2018b), CIFAR-FS (Bertinetto et al., 2019) and FC-100 (Oreshkin et al., 2018). In all experiments, we follow the standard data usage specifications same as (Hiller et al., 2022), splitting data into the meta-training set, meta-validation set, and meta-test set, and classes in each set are mutually exclusive. The details of each dataset are described in Appendix L.1.

**Implementation Details.** We train our method in two stages following Hiller et al. (2022): self-supervised pretraining and meta-tuning. We first pre-train our vision transformer backbone (Dosovitskiy et al., 2020; Liu et al., 2021) utilizing a self-supervised training objective (Zhou et al., 2022a). Subsequently, we integrate our proposed SAM and TAM into the original vision transformer for meta-learning. We denote MetaFormer-I to predict queries independently in the inductive setting and MetaFormer-A for the autoregressive scenario. Further detailed training and evaluation settings are included in the Appendix L.2.

**Comparison to the State-Of-The-Art Methods.** The comparison results with related or recent state-of-the-art (SOTA) methods on *mini*ImageNet and *tiered*ImageNet is shown in Table 1. In comparison to previous state-of-the-art meta-learning approaches, our method outperforms them significantly. For example, on *mini*ImageNet, MetaFormer-I surpasses its meta-learning competitor FewTURE (Hiller et al., 2022) by 7.76% and 5.51% in 1-shot and 5-shot settings, respectively. This demonstrates the remarkable effectiveness of our proposed holistic attention mechanism to fully leverage the transformer potential for meta-learning. SMKD+MetaFormer-I also outperforms self-distillation based methods (He et al., 2022b; Lin et al., 2023). Table 2 displays results on the CIFAR-FS and FC100 datasets. MetaFormer-I also achieves better performance than previous methods, which shows the superiority of our proposed method. We note that our MetaFormer-A significantly enhances performance, establishing a new baseline for autoregressive few-shot image classification tasks. See Table 4 in Appendix C for the comparison with Swin backbone.

Table 1: Average classification accuracy (%) for 5-way 1-shot and 5-way 5-shot scenarios. Reported are the mean and 95% confidence interval on the unseen test sets of *mini*ImageNet (Vinyals et al., 2016a) and *tiered*ImageNet (Ren et al., 2018a), using the established evaluation protocols.

| Method | Backbone | # Params | *mini*ImageNet | | *tiered*ImageNet | |
| --- | --- | --- | --- | --- | --- | --- |
| | | | 1-shot | 5-shot | 1-shot | 5-shot |
| MatchNet (Vinyals et al., 2016b) | *ResNet-12* | 12.4 M | $61.24_{\pm0.29}$ | $73.93_{\pm0.23}$ | $71.01_{\pm0.33}$ | $83.12_{\pm0.24}$ |
| ProtoNet (Snell et al., 2017) | *ResNet-12* | 12.4 M | $62.29_{\pm0.33}$ | $79.46_{\pm0.48}$ | $68.25_{\pm0.23}$ | $84.01_{\pm0.56}$ |
| FEAT (Ye et al., 2020) | *ResNet-12* | 14.1 M | $66.78_{\pm0.20}$ | $82.05_{\pm0.14}$ | $70.80_{\pm0.23}$ | $84.79_{\pm0.16}$ |
| DeepEMD (Zhang et al., 2020a) | *ResNet-12* | 12.4 M | $65.91_{\pm0.82}$ | $82.41_{\pm0.56}$ | $71.16_{\pm0.87}$ | $86.03_{\pm0.58}$ |
| IEPT (Zhang et al., 2020b) | *ResNet-12* | 12.4 M | $67.05_{\pm0.44}$ | $82.90_{\pm0.30}$ | $72.24_{\pm0.50}$ | $86.73_{\pm0.34}$ |
| MELR (Fei et al., 2020) | *ResNet-12* | 14.1 M | $67.40_{\pm0.43}$ | $83.40_{\pm0.28}$ | $72.14_{\pm0.51}$ | $87.01_{\pm0.35}$ |
| FRN (Wertheimer et al., 2021) | *ResNet-12* | 12.4 M | $66.45_{\pm0.19}$ | $82.83_{\pm0.13}$ | $72.06_{\pm0.22}$ | $86.89_{\pm0.14}$ |
| CG (Zhao et al., 2021) | *ResNet-12* | 12.4 M | $67.02_{\pm0.20}$ | $82.32_{\pm0.14}$ | $71.66_{\pm0.23}$ | $85.50_{\pm0.15}$ |
| DMF (Xu et al., 2021) | *ResNet-12* | 12.4 M | $67.76_{\pm0.46}$ | $82.71_{\pm0.31}$ | $71.89_{\pm0.52}$ | $85.96_{\pm0.35}$ |
| BML (Zhou et al., 2021b) | *ResNet-12* | 12.4 M | $67.04_{\pm0.63}$ | $83.63_{\pm0.29}$ | $68.99_{\pm0.50}$ | $85.49_{\pm0.34}$ |
| CNL (Zhao et al., 2021) | *ResNet-12* | 12.4 M | $67.96_{\pm0.98}$ | $83.36_{\pm0.51}$ | $73.42_{\pm0.95}$ | $87.72_{\pm0.75}$ |
| Meta-NVG (Zhang et al., 2021a) | *ResNet-12* | 12.4 M | $67.14_{\pm0.80}$ | $83.82_{\pm0.51}$ | $74.58_{\pm0.88}$ | $86.73_{\pm0.61}$ |
| RENet (Kang et al., 2021) | *ResNet-12* | 12.6 M | $67.60_{\pm0.44}$ | $82.58_{\pm0.30}$ | $71.61_{\pm0.51}$ | $85.28_{\pm0.35}$ |
| PAL (Ma et al., 2021a) | *ResNet-12* | 12.4 M | $69.37_{\pm0.64}$ | $84.40_{\pm0.44}$ | $72.25_{\pm0.72}$ | $86.95_{\pm0.47}$ |
| COSOC (Luo et al., 2021) | *ResNet-12* | 12.4 M | $69.28_{\pm0.49}$ | $85.16_{\pm0.42}$ | $73.57_{\pm0.43}$ | $87.57_{\pm0.10}$ |
| Meta DeepBDC (Xie et al., 2022) | *ResNet-12* | 12.4 M | $67.34_{\pm0.43}$ | $84.46_{\pm0.28}$ | $72.34_{\pm0.49}$ | $87.31_{\pm0.32}$ |
| LEO (Rusu et al., 2018) | *WRN-28-10* | 36.8 M | $61.76_{\pm0.08}$ | $77.59_{\pm0.12}$ | $66.33_{\pm0.05}$ | $81.44_{\pm0.09}$ |
| MetaFun (Xu et al., 2020) | *WRN-28-10* | 37.7 M | $62.12_{\pm0.30}$ | $78.20_{\pm0.16}$ | $67.72_{\pm0.14}$ | $83.28_{\pm0.12}$ |
| CC+rot (Gidaris et al., 2019) | *WRN-28-10* | 36.5 M | $62.93_{\pm0.45}$ | $79.87_{\pm0.33}$ | $70.53_{\pm0.51}$ | $84.98_{\pm0.36}$ |
| FEAT (Ye et al., 2020) | *WRN-28-10* | 38.1 M | $65.10_{\pm0.20}$ | $81.11_{\pm0.14}$ | $70.41_{\pm0.23}$ | $84.38_{\pm0.16}$ |
| MetaQDA (Zhang et al., 2021c) | *WRN-28-10* | 36.5 M | $67.83_{\pm0.64}$ | $84.28_{\pm0.69}$ | $74.33_{\pm0.65}$ | $89.56_{\pm0.79}$ |
| OM (Qi et al., 2021) | *WRN-28-10* | 36.5 M | $66.78_{\pm0.30}$ | $85.29_{\pm0.41}$ | $71.54_{\pm0.29}$ | $87.79_{\pm0.46}$ |
| SUN (Dong et al., 2022) | *ViT* | 12.5 M | $67.80_{\pm0.45}$ | $83.25_{\pm0.30}$ | $72.99_{\pm0.50}$ | $86.74_{\pm0.33}$ |
| FewTURE (Hiller et al., 2022) | *ViT-Small* | 22 M | $68.02_{\pm0.88}$ | $84.51_{\pm0.53}$ | $72.96_{\pm0.92}$ | $86.43_{\pm0.67}$ |
| FewTURE (Hiller et al., 2022) | *Swin-Tiny* | 29 M | $72.40_{\pm0.78}$ | $86.38_{\pm0.49}$ | $76.32_{\pm0.87}$ | $89.96_{\pm0.55}$ |
| MetaFormer-I (Ours) | *ViT-Small* | 24.5 M | $\mathbf{75.78}_{\pm\mathbf{0.71}}$ | $\mathbf{90.02}_{\pm\mathbf{0.44}}$ | $\mathbf{79.05}_{\pm\mathbf{0.81}}$ | $\mathbf{90.40}_{\pm\mathbf{0.53}}$ |
| MetaFormer-A (Ours) | *ViT-Small* | 24.5 M | $\mathbf{79.41}_{\pm\mathbf{0.73}}$ | $\mathbf{91.21}_{\pm\mathbf{0.44}}$ | $\mathbf{84.41}_{\pm\mathbf{0.79}}$ | $\mathbf{92.47}_{\pm\mathbf{0.47}}$ |
| HCTransformers (He et al., 2022b) | $3\times$*ViT-Small* | 63 M | $74.74_{\pm0.17}$ | $89.19_{\pm0.13}$ | $79.67_{\pm0.20}$ | $91.72_{\pm0.11}$ |
| SMKD (Lin et al., 2023) | *ViT-Small* | 21 M | $74.28_{\pm0.18}$ | $88.89_{\pm0.09}$ | $78.83_{\pm0.20}$ | $91.21_{\pm0.11}$ |
| SMKD + MetaFormer-I (Ours) | *ViT-Small* | 24.5 M | $\mathbf{76.54}_{\pm\mathbf{0.73}}$ | $\mathbf{90.76}_{\pm\mathbf{0.41}}$ | $\mathbf{80.57}_{\pm\mathbf{0.82}}$ | $\mathbf{92.42}_{\pm\mathbf{0.49}}$ |
| SMKD + MetaFormer-A (Ours) | *ViT-Small* | 24.5 M | $\mathbf{81.61}_{\pm\mathbf{0.75}}$ | $\mathbf{92.25}_{\pm\mathbf{0.40}}$ | $\mathbf{84.43}_{\pm\mathbf{0.80}}$ | $\mathbf{93.41}_{\pm\mathbf{0.49}}$ |

Table 2: Average classification accuracy (%) for 5-way 1-shot and 5-way 5-shot scenarios. Reported are the mean and 95% confidence interval on the unseen test sets of CIFAR-FS (Bertinetto et al., 2019) and FC100 (Oreshkin et al., 2018), using the established evaluation protocols.

| Method | Backbone | # Params | CIFAR-FS | | FC100 | |
|---|---|---|---|---|---|---|
| | | | 1-shot | 5-shot | 1-shot | 5-shot |
| ProtoNet (Snell et al., 2017) | ResNet-12 | 12.4 M | - | - | $41.54_{\pm0.76}$ | $57.08_{\pm0.76}$ |
| MetaOpt (Lee et al., 2019) | ResNet-12 | 12.4 M | $72.00_{\pm0.70}$ | $84.20_{\pm0.50}$ | $41.10_{\pm0.60}$ | $55.50_{\pm0.60}$ |
| MABAS (Kim et al., 2020) | ResNet-12 | 12.4 M | $73.51_{\pm0.92}$ | $85.65_{\pm0.65}$ | $42.31_{\pm0.75}$ | $58.16_{\pm0.78}$ |
| RFS (Tian et al., 2020) | ResNet-12 | 12.4 M | $73.90_{\pm0.80}$ | $86.90_{\pm0.50}$ | $44.60_{\pm0.70}$ | $60.90_{\pm0.60}$ |
| Meta-NVG (Zhang et al., 2021a) | ResNet-12 | 12.4 M | $74.63_{\pm0.91}$ | $86.45_{\pm0.59}$ | $46.40_{\pm0.81}$ | $61.33_{\pm0.71}$ |
| RENet (Kang et al., 2021) | ResNet-12 | 12.6 M | $74.51_{\pm0.46}$ | $86.60_{\pm0.32}$ | - | - |
| TPMN (Wu et al., 2021) | ResNet-12 | 12.4 M | $75.50_{\pm0.90}$ | $87.20_{\pm0.60}$ | $46.93_{\pm0.71}$ | $63.26_{\pm0.74}$ |
| MixFSL (Afrasiyabi et al., 2021) | ResNet-12 | 12.4 M | - | - | $44.89_{\pm0.63}$ | $60.70_{\pm0.60}$ |
| PSST (Chen et al., 2021b) | WRN-28-10 | 36.5 M | $77.02_{\pm0.38}$ | $88.45_{\pm0.35}$ | - | - |
| Meta-QDA (Zhang et al., 2021c) | WRN-28-10 | 36.5 M | $75.83_{\pm0.88}$ | $88.79_{\pm0.75}$ | - | - |
| SUN (Dong et al., 2022) | ViT | 12.5M | $78.37_{\pm0.46}$ | $88.84_{\pm0.32}$ | - | - |
| FewTURE (Hiller et al., 2022) | ViT-Small | 22 M | $76.10_{\pm0.88}$ | $86.14_{\pm0.64}$ | $46.20_{\pm0.79}$ | $63.14_{\pm0.73}$ |
| FewTURE (Hiller et al., 2022) | Swin-Tiny | 29 M | $77.76_{\pm0.81}$ | $88.90_{\pm0.59}$ | $47.68_{\pm0.78}$ | $63.81_{\pm0.75}$ |
| MetaFormer-I (Ours) | ViT-Small | 24.5 M | $\mathbf{80.16_{\pm0.76}}$ | $\mathbf{90.57_{\pm0.55}}$ | $\mathbf{51.14_{\pm0.71}}$ | $\mathbf{68.33_{\pm0.74}}$ |
| MetaFormer-A (Ours) | ViT-Small | 24.5 M | $\mathbf{83.48_{\pm0.75}}$ | $\mathbf{91.62_{\pm0.53}}$ | $\mathbf{53.76_{\pm0.80}}$ | $\mathbf{70.68_{\pm0.74}}$ |
| HCTransformers (He et al., 2022b) | $3\times$ViT-Small | 63 M | $78.89_{\pm0.18}$ | $90.50_{\pm0.09}$ | $48.27_{\pm0.15}$ | $66.42_{\pm0.16}$ |
| SMKD (Lin et al., 2023) | ViT-Small | 21M | $80.08_{\pm0.18}$ | $90.91_{\pm0.13}$ | $50.38_{\pm0.16}$ | $68.50_{\pm0.16}$ |
| SMKD + MetaFormer-I (Ours) | ViT-Small | 24.5 M | $\mathbf{81.49_{\pm0.74}}$ | $\mathbf{91.91_{\pm0.54}}$ | $\mathbf{52.18_{\pm0.78}}$ | $\mathbf{71.29_{\pm0.73}}$ |
| SMKD + MetaFormer-A (Ours) | ViT-Small | 24.5 M | $\mathbf{85.59_{\pm0.76}}$ | $\mathbf{92.85_{\pm0.54}}$ | $\mathbf{55.68_{\pm0.86}}$ | $\mathbf{73.31_{\pm0.77}}$ |

## 4.2 BROADER STUDY OF FEW-SHOT LEARNING

To further investigate the fast adaptation ability of our method, we evaluate the MetaFormer in more challenging cross-domain (Chen et al., 2019; Oh et al., 2022) and multi-domain (Triantafillou et al., 2020) scenarios, containing both the class and domain shifts. Appendix M and Appendix N provide the benchmark datasets and implementation details.

**Cross-Domain and Multi-Domain Few-shot Classification Results.** We evaluate MetaFomer meta-trained on *mini*ImageNet on cross-domain few-shot classification benchmarks following Oh et al. (2022) in Table 12 (Appendix M.3). Compared with previous in-domain state-of-the-art meta-learning (Hiller et al., 2022) and self-supervised learning (Lin et al., 2023) methods, MetaFormer achieves significant performance improvement by up to $10.51\%$, underscoring its task adaptability in the face of domain gaps. In Table 13 (Appendix N.3), we assess the effectiveness of MetaFormer on the large-scale and challenging Meta-Dataset. MetaFormer surpasses PMF (Hu et al., 2022) to handle tasks with substantially different distributions. We attribute such impressive improvement to our proposed holistic attention, a mechanism that not only facilitates sample correspondence learning but also enables knowledge reuse through inter-task attention, thus aiding task adaptation to obtain more discriminative feature representations in each task.

## 4.3 ABLATION STUDY

Table 3: Component ablation studies and the number of additional learnable parameters on *mini*ImageNet.

| SAM | TAM | Add. Params. | *mini*ImageNet | |
|---|---|---|---|---|
| | | | 1-shot | 5-shot |
| ✓ | ✓ | +3.57M | $75.78 \pm 0.71$ | $90.02 \pm 0.44$ |
| ✓ | ✗ | +2.01M | $74.64 \pm 0.76$ | $87.53 \pm 0.47$ |
| ✗ | ✓ | +1.56M | $73.63 \pm 0.75$ | $87.76 \pm 0.52$ |

**Component Analysis.** In this section, we investigate the individual contributions of each component in MetaFormer by removing the components one by one: Sample-level Attention Module (SAM), and Task-level Attention Module (TAM). The impact on performance and the increase in the number of additional learnable parameters are detailed in Table 3. This table validates the contribution of each module, demonstrating that they enhance performance with only a modest increase in computational overhead. Specifically, the introduction of SAM results in a $2.15\%$ performance gain in the 1-shot

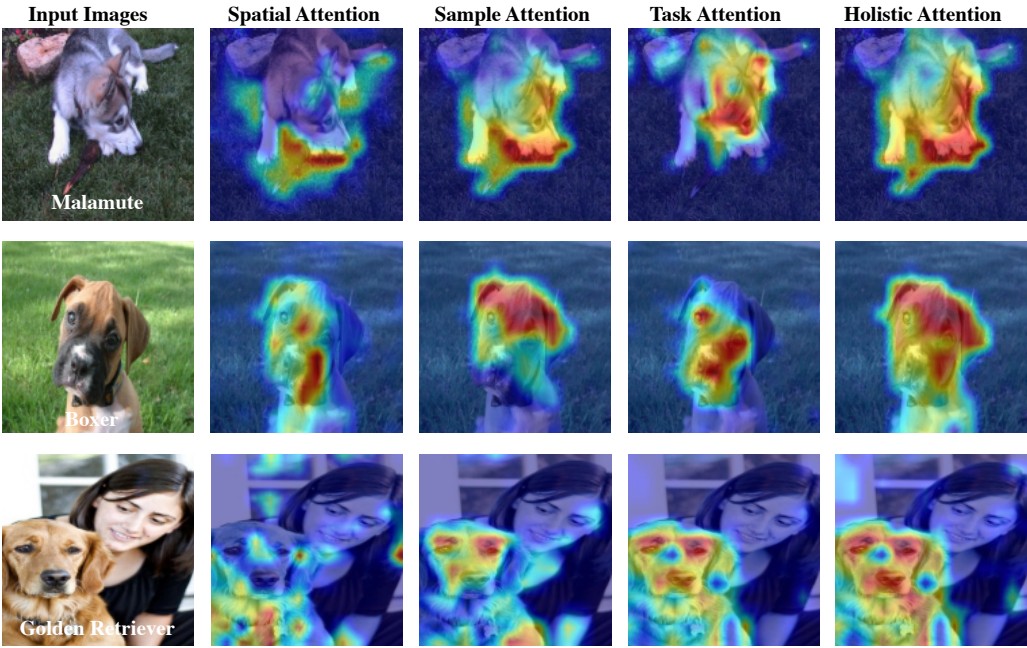

Figure 3: Response visualization for MetaFormer with holistic attention.

setting by facilitating sample correspondence learning, thereby enhancing consistency within the task. Furthermore, the incorporation of TAM further leads to an additional improvement of $2.49\%$ in the 5-shot setting, is achieved by regularizing the current task with retrieved relevant semantics. See Appendix D for more ablation analysis.

## 4.4 QUALITATIVE ANALYSIS

Figure 3 shows visualizations of our holistic attention. Columns respectively illustrate the attention map of three attention modules. The results demonstrate that the sample correspondence learning guided by spatial attention and sample attention modules can suppress irrelevant regions via exploiting pattern relations within and across samples, thereby learning more discriminative task-specific features. Building on this foundation, the task attention module facilitates the transfer of semantic knowledge from the previous task to the new one, focusing particularly on the key components of foreground objects. When integrated with intra- and inter-task attention, our holistic attention yields a more accurate and comprehensive response map concentrated on the foreground region.

## 5 CONCLUSIONS

This paper proposes MetaFormer, a novel Vit-backed meta-learning approach for few-shot classification. To fully leverage transformer characteristics, MetaFormer builds holistic attention by introducing two lightweight modules to capture intra-task and inter-task interactions. With the Sample-level Attention Module (SAM), MetaFormer captures task-specific discriminative feature representations by facilitating sample correspondence learning to enforce consistency within a task. Meanwhile, the Task-level Attention Module (TAM) retrieves most relevant knowledge from seen tasks to regularize learning of the current task via maintaining a dynamic knowledge pool. We also extend our MetaFormer to build the new baseline in the autoregressive few-shot image classification setting. Extensive experiments demonstrate the superiority of MetaFormer in the meta-learning approach family, achieving remarkable performance on the standard in-domain benchmarks as well as more challegning cross-domain and multi-domain benchmarks.

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

# A    MORE QUALITATIVE RESULTS

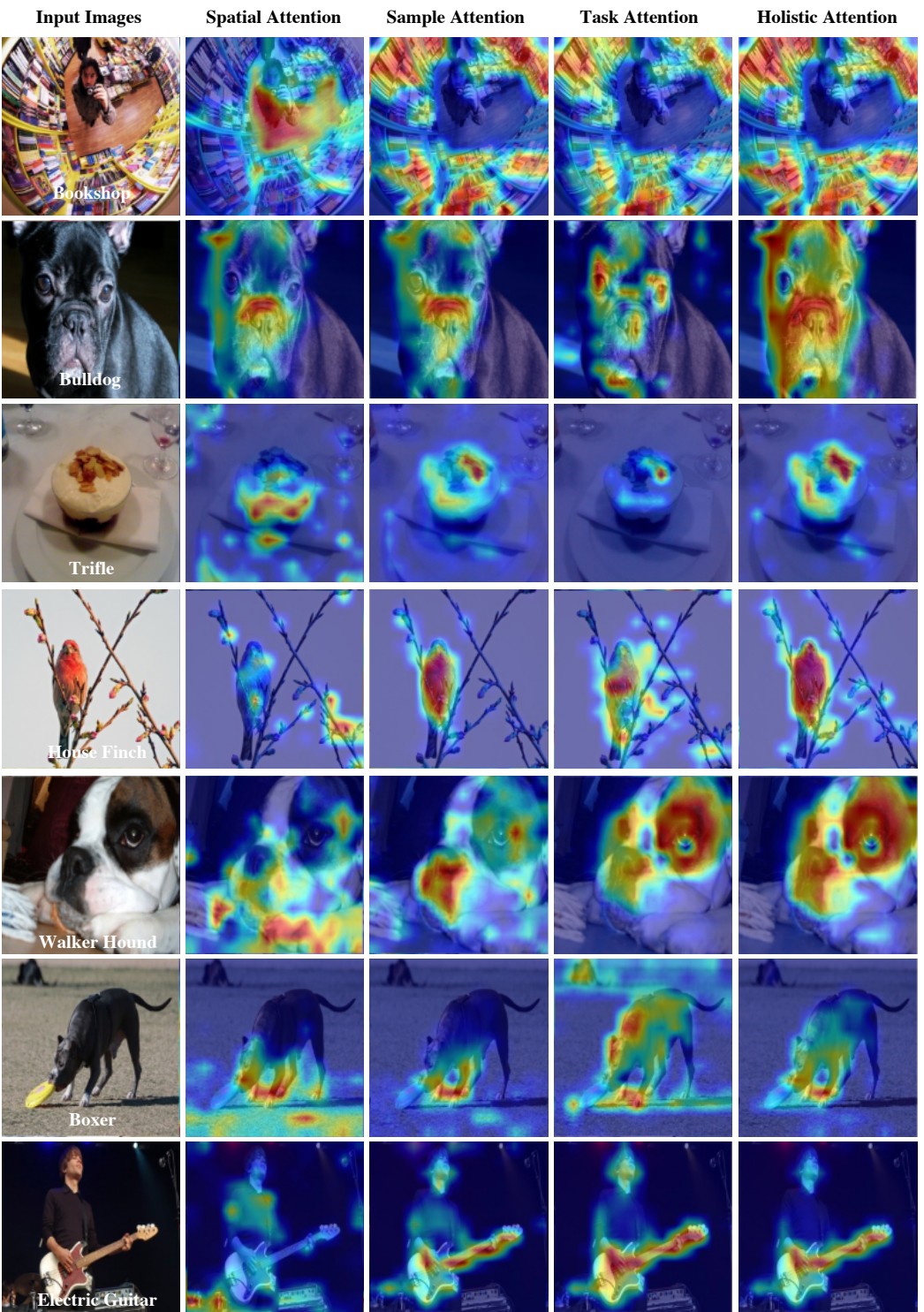

Figure 4: More visualization of our holistic attention mechanism. Three attention modules collaboratively focusing on task-specific foreground regions.

## B  ILLUSTRATION OF OUR DECOUPLED SPACE-SAMPLE ATTENTION

Figure 5 compares the complexity between the joint space-sample attention and our decoupled space-sample attention.

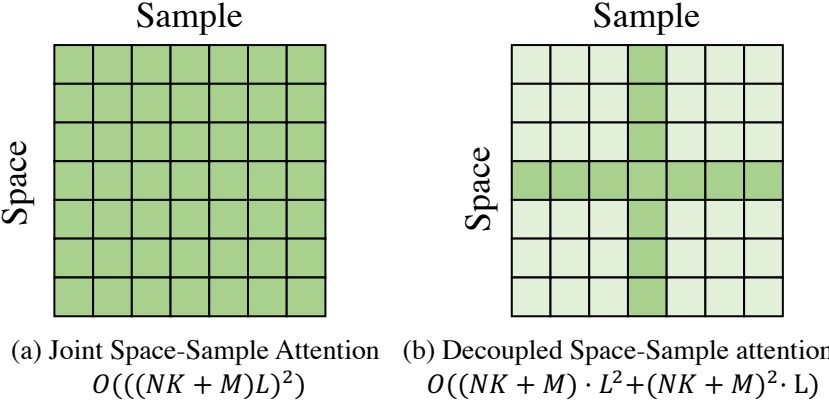

(a) Joint Space-Sample Attention
$O(((NK + M)L)^2)$

(b) Decoupled Space-Sample attention
$O((NK + M) \cdot L^2 + (NK + M)^2 \cdot L)$

Figure 5: Complexity comparison with joint space-sample attention approaches. For a $N$-way $K$-shot task with $M$ queries, our method decouples the space-sample attention by first performing self-attention between $L$ patches within each image to aggregate spatial information and then computing sample interactions across all patches at the same spatial location to capture the similarities and variances among samples.

## C  METAFORMER ON HIERARCHICAL TRANSFORMERS.

We extend MetaFormer to Swin (Liu et al., 2021) which employs shifted local window attention for performing self-attention inside each window and merges patch embeddings to build hierarchical structures. In Table 4, the experiments are conducted on the same pre-trained Swin-Tiny and our MetaFormer continues to outperform FewTURE for both settings.

Table 4: Comparison results with the Swin-Transformer backbone on *mini*Imagenet.

| Method | BackBone | 1-shot | 5-shot |
|---|---|---|---|
| FewTURE (Hiller et al., 2022) | *Swin-Tiny* | 72.40±0.78 | 86.38±0.49 |
| MetaFormer-I (Ours) | *Swin-Tiny* | **74.17**±**0.73** | **89.17**±**0.45** |

## D  ABALTION OF OTHER DESIGN STRATEGIES.

Table 5: Comparisons of different architecture design strategies. We report 1-shot accuracy on *mini*ImageNet for different choices.

(a) Different label infusion methods.

| Method | Acc. (%) |
|---|---|
| w/o. label | 73.70 |
| concatenation | 74.59 |
| summation | **74.64** |

(b) Different SAM locations.

| Location | Acc. (%) |
|---|---|
| $[5, 7, 9]$ | 74.36 |
| $[6, 8, 10]$ | **74.64** |
| $[11]$ | 72.34 |

(c) Different SAM variants.

| Method | Acc. (%) |
|---|---|
| within-support | 73.30 |
| support-query | 73.28 |
| global features | 71.49 |

In Table 5, we explore various design choices for SAM. The results in Table 5a emphasize the significance of label infusion for support patch tokens before sample correspondence learning.

Interestingly, our investigation reveals that the choice of label infusion strategy, concatenating one-hot labels or utilizing summation via linear projection, has a minimal effect on performance.

The variability of feature semantics across different layers in the backbone leads us to investigate which layer optimally facilitates sample interaction. The results are shown in Table 5b, indicating that starting to build intra-task interaction from stage 6 is moderate. Notably, the integration of multi-scale semantic information accounts for an improvement of 2.3%.

Figure 6 shows the two alternative sample causal masks for our MetaFormer-I, where we encode the sample relationship separately. As shown in Table 5c, the ablated masks of within-support and support-query manifest sub-optimal performance, further validating that SAM with the inductive mask works not because of the introduction of extra parameters. This also underscores the complementary benefits of employing both interactions in our design on enhanced task-specific representations. Additionally, we observe that relying solely on global image features incurs a significant loss of critical information necessary for capturing discriminative relationships among samples.

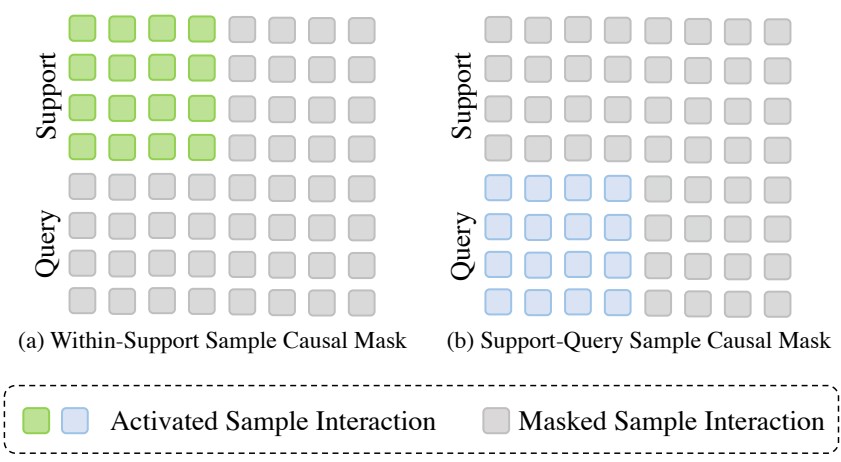

(a) Within-Support Sample Causal Mask    (b) Support-Query Sample Causal Mask

Activated Sample Interaction    Masked Sample Interaction

Figure 6: The alternative sample casual masks for MetaFormer-I.

# E    INFERENCE TIME COMPARISION.

We have conducted a detailed comparative analysis of the computational efficiency between our MetaFormer and other state-of-the-art methods, as presented in Table 6. We test by an NVIDIA RTX A6000 GPU and report the performance and inference latency for 5-way 1-shot and 5-way 5-shot on miniImageNet. In our analysis, we consider the inference-time tuning cost for FewTURE. The results indicate that our MetaFomer demonstrates computational efficiency over other compared methods. Importantly, our method outpaces previous meta-learning counterpart FewTURE in both speed and performance, primarily due to eliminating the need for inference-time tuning. It is noteworthy that MetaFormer facilitates full sample interactions, as opposed to FewTURE, which only considers the contextual relationships.

# F    THE NUMBER OF PARAMETERS COMPARISION.

In Table 7, we present a selection of representative and state-of-the-art methods, detailing their number of backbone parameters and the total number of parameters. Simply increasing more parameters by changing the backbone architecture does not necessarily lead to better performance. We observe that FewTURE (Hiller et al., 2022) and HCTransformers (He et al., 2022b), despite possessing a larger number of parameters, markedly lags behind the proposed MetaFormer. We also conduct a comparative analysis with an ablated version, achieved by naively augmenting the number of layers in

Table 6: Comparison of average classification accuracy and inference times on the *mini*-ImageNet test set.

| Setting | Method | Accuracy | GLOPs | Inference time [ms] |
|---|---|---|---|---|
| 1-shot | FewTURE (Hiller et al., 2022) | $68.02_{\pm 0.88}$ | 5.01 | $77.35_{\pm 0.47}$ |
| | SMKD (Lin et al., 2023) | $74.28_{\pm 0.18}$ | 12.58 | $137.58_{\pm 0.66}$ |
| | MetaFormer-I | $\mathbf{75.78}_{\pm 0.71}$ | 4.88 | $\mathbf{67.65}_{\pm 0.78}$ |
| 5-shot | FewTURE (Hiller et al., 2022) | $84.51_{\pm 0.53}$ | 5.01 | $111.22_{\pm 1.27}$ |
| | SMKD (Lin et al., 2023) | $88.82_{\pm 0.09}$ | 12.58 | $171.37_{\pm 0.78}$ |
| | MetaFormer-I | $\mathbf{90.02}_{\pm 0.44}$ | 4.88 | $\mathbf{105.72}_{\pm 1.06}$ |

ViT-Small to make it comparable with the proposed MetaFormer. The results presented substantiate that merely increasing parameters cannot fully address the challenges inherent in few-shot learning. In fact, such augmentation may even elevate the risk of overfitting. It's crucial to demonstrate that our enhancements are not merely due to an increased parameter count. We think that our proposed MetaFormer is cost-effective considering the remarkable performance gains over previous meta-learning SOTA FewTURE (Hiller et al., 2022) with a modest increase of additional parameters. Also, note that the conversion from inductive to autoregressive version leads to no extra parameters, further emphasizing its efficiency.

Table 7: Comparsion of state-of-the-art methods with the number of parameters.

| Method | Backbone | $\approx$ # Params | # Total Params | *mini*ImageNet 1-shot | 5-shot |
|---|---|---|---|---|---|
| FEAT (Ye et al., 2020) | *ResNet-12* | 12.4 M | 14.1 M | $66.78_{\pm 0.20}$ | $82.05_{\pm 0.14}$ |
| FEAT (Ye et al., 2020) | *WRN-28-10* | 36.5 M | 38.1 M | $65.10_{\pm 0.20}$ | $81.11_{\pm 0.14}$ |
| FewTURE (Hiller et al., 2022) | *ViT-Small* | 21 M | 22 M | $68.02_{\pm 0.88}$ | $84.51_{\pm 0.53}$ |
| FewTURE (Hiller et al., 2022) | *Swin-Tiny* | 28 M | 29 M | $72.40_{\pm 0.78}$ | $86.38_{\pm 0.49}$ |
| HCTransformers (He et al., 2022b) | $3\times$*ViT-Small* | 63 M | 63 M | $74.74_{\pm 0.17}$ | $89.19_{\pm 0.13}$ |
| SMKD (Lin et al., 2023) | *ViT-Small* | 21 M | 21 M | $74.28_{\pm 0.18}$ | $88.89_{\pm 0.09}$ |
| ViT with more layers | *ViT-Small* | 21 M | 25.2 M | $69.75_{\pm 0.71}$ | $84.12_{\pm 0.56}$ |
| MetaFormer-I (Ours) | *ViT-Small* | 21 M | 24.5 M | $\mathbf{75.78}_{\pm 0.71}$ | $\mathbf{90.02}_{\pm 0.44}$ |
| MetaFormer-A (Ours) | *ViT-Small* | 21 M | 24.5 M | $\mathbf{79.41}_{\pm 0.73}$ | $\mathbf{91.21}_{\pm 0.44}$ |

# G    COMPARISON WITH OTHER INTER-TASK ATTENTION METHODS

In this section, we compare our TAM with the inter-task attention module (IT-att) as presented in Wang et al. (2022a). We note that both problem settings and motivations for these modules are distinct. TAM is rooted in the domain of few-shot learning, where the paramount concern is facilitating knowledge transfer between tasks. In contrast, IT-att in Wang et al. (2022a) is grounded in continual learning, where the primary focus lies in mitigating catastrophic forgetting. While both TAM and IT-att seemingly adopt a learnable embedding for each task, TAM utilizes it to represent the knowledge specific to the current task. In contrast, IT-att stores all past knowledge in it through regularization-based consolidation mentioned below. Owing to disparate problem settings, TAM maintains a knowledge pool that stores an array of task-dependent embeddings. In contrast, IT-att keeps a record of a single key and a single bias. Leveraging our knowledge pool, we consolidate the current task probe vector by averaging it with the most relevant vector in the pool (refer to equation 5). In contrast, IT-att, which is designed to address forgetting, employs importance-based regularization to enforce proximity of the current task embedding and previous one. And thus the task interaction in TAM exhibits greater flexibility and expressiveness, aligning more closely with the objective of knowledge transfer in few-shot learning. We conducted an ablation study on *mini*ImageNet for the empirical comparison in Table 8, wherein we implement IT-att in our setting. The results showcase that our proposed TAM consistently outperforms IT-att by approximately 0.6% on 1-shot and 1.3% on 5-shot settings.

Table 8: Comparison results with the inter-task attention module (Wang et al., 2022a) on *mini*Imagenet.

| Method | BackBone | 5-shot |
|--------|----------|--------|
| IT-att | *ViT-Small* | $88.70_{\pm 0.50}$ |
| TAM | *ViT-Small* | $\mathbf{90.02}_{\pm \mathbf{0.44}}$ |

## H COMPARISON WITH CNN-BASED META-LEARNING METHODS

In this section, we give more in-depth discussions concerning prior research in the realm of meta-learning that incorporates vision transformers as their foundation architectures. We posit that the challenge of architectural inconsistency partially accounts for the limited research in the realm of meta-learning grounded on ViT. In the Table 9, we adapt FiLM, a technique commonly employed in CNN-based meta-learning for task adaptation through conditioned batch normalization (Requeima et al., 2019; Oreshkin et al., 2018), into layer normalization layers of ViT for task conditioning. As shown in the table, our experiments reveal a performance drop when ViT was applied with FiLM. Another key challenge is the increased parameter requirement of ViT. FewTURE (Hiller et al., 2022), as expounded in the Related Work section, is the pioneering work that tailors to ViT via inner-loop token importance reweighting, and addresses the second challenge via self-supervised pre-training on the meta-training dataset. Our approach, empowering sample-to-sample and task-to-task interaction, further improves the accuracy substantially.

Table 9: Comparison results with different meta-learning approaches for Vision Transformer on the *mini*Imagenet.

| Method | BackBone | 5-way 1-shot |
|--------|----------|--------------|
| Vanilla ViT | *ViT-Small* | $69.03_{\pm 0.71}$ |
| ViT+FiLM | *ViT-Small* | $58.75_{\pm 0.73}$ |
| MetaFormer-I | *ViT-Small* | $\mathbf{75.78}_{\pm \mathbf{0.71}}$ |
| MetaFormer-I | *ViT-Small* | $\mathbf{79.41}_{\pm \mathbf{0.73}}$ |

## I STUDY ON LARGE-SCALE FOUNDATION MODELS

Pre-trained vision foundation models demonstrate impressive zero-shot image classification performance (Radford et al., 2021). Recent work have shown that CLIP's performance on downstream tasks can be further enhanced by utilizing few-shot data and techniques (Zhang et al., 2021b; Zhou et al., 2022b; Zhu et al., 2023). These approaches have shown promising improvements over frozen models like Zero-shot CLIP. To further investigate the adaptation ability of our proposed method, we adapt our method to CLIP model with ViT-B/16 for advancing its performance in downstream tasks.

### I.1 IMPLEMENTATION DETAILS

We evaluate different methods on 1-shot EuroSAT (Helber et al., 2019) and ISIC (Tschandl et al., 2018) datasets. For both training and testing phases, we employ the episodic approach as described in Vinyals et al. (2016b). Note that we strictly follow the TiP-Adapter-F (Zhang et al., 2021b) pipeline to sample support set from the train set and query set from the test set to build the task for evaluation, since there are no new classes in the test set. For instance, in the EuroSAT dataset with 10 classes, we construct the 10-way 1-shot task, where the support and query sets are drawn from the train and test split of the EuroSAT dataset, respectively. We also integrate the cache model from TiP-Adapter as the auxiliary classifier head. We only fine-tune introduced SAM modules to and keep frozen the visual encoder and textual encoder of CLIP. We train our method for 20 epochs on both datasets and we employ the SGD optimizer with a cosine-decaying learning rate initiated at $2 \times 10^{-4}$, a momentum value of 0.9, and a weight decay of $5 \times 10^{-4}$. We test using the pre-trained word embeddings of a single prompt, "a photo of a [CLASS]." for all methods.

## I.2 RESULTS

As shown in the Table 10, the CLIP pre-trained on large-scale web-crawled image-text pairs struggles with downstream datasets exhibiting a large domain gap, such as the medical dataset of ISIC. Adapting the CLIP with a downstream dataset is pivotal to guarantee better performance, though naively increasing the number of parameters to adapt even incurs overfitting. Our method significantly enhances Zero-shot CLIP on EuroSAT by 42.76% and ISIC by 43.81%, and its adaptation ability also surpasses Tip-Adapter by a large margin.

Table 10: Classification accuracy (%) for 1-shot EuroSAT and ISIC. Reported are the mean and 95% confidence interval on the test set. ViT-B/16 with the patch size $16 \times 16$ is adopted for the vision branch in all methods.

| Method | EuroSAT | ISIC |
|---|---|---|
| Zero-shot CLIP | $48.73_{\pm 0.98}$ | $21.07_{\pm 0.76}$ |
| Tip-Adapter | $69.85_{\pm 0.75}$ | $28.70_{\pm 0.97}$ |
| Tip-Adapter-F | $72.01_{\pm 0.97}$ | $32.27_{\pm 1.11}$ |
| Tip-Adapter-F with more layers | $51.95_{\pm 0.86}$ | $16.17_{\pm 0.78}$ |
| Tip-Adapter+MetaFormer (Ours) | $\mathbf{88.83_{\pm 0.78}}$ | $\mathbf{45.96_{\pm 1.44}}$ |

## J ADDITIONAL ANALYSIS OF TASK PROBE VECTOR

As shown in Figure 7, we analyze the task probe vectors on *mini*ImageNet across different tasks sampled from meta-train and meta-test sets. The visualization effectively underscores the efficacy of the learned task probe vectors in capturing task relationships. For example, we observe a higher similarity in task features among tasks involving car tires, dogs, and long-legged animals. This demonstrates MetaFormer's capability in discerning and utilizing task dynamics.

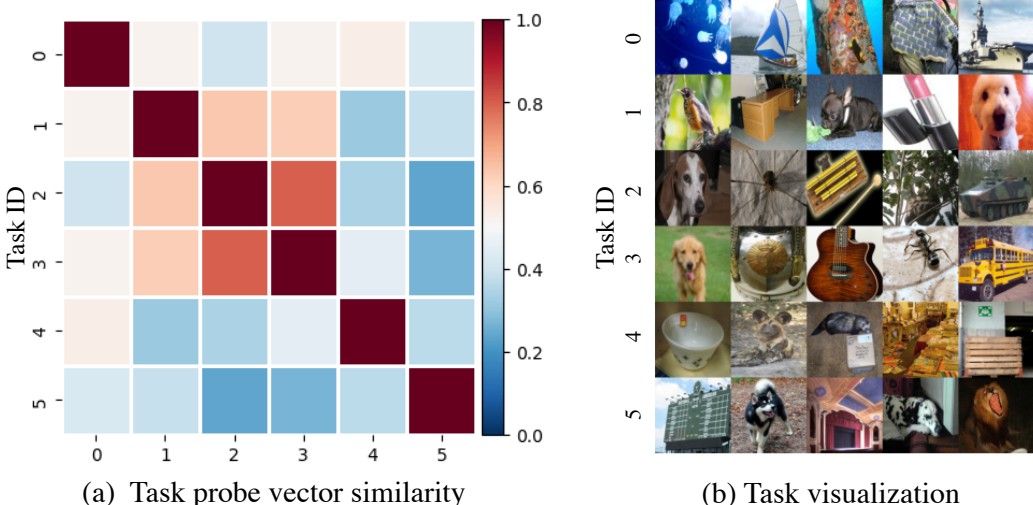

(a) Task probe vector similarity     (b) Task visualization

Figure 7: Interpretation of task probe vector. Each task is randomly selected from *mini*ImageNet. (a) We show the similarity heatmap between task probe vectors, where deeper color means higher similarity. (b) We show the visualization of the corresponding tasks.

## K  COMPARISON WITH TRANSDUCTIVE METHODS

In this section, we conduct additional comparisons for MetaFormer-A against other state-of-the-art transductive methods in Table 11. Under the transductive setting, the proposed MetaFormer-A by only setting the autoregressive sample causal mask requires only one singe-pass during inference, which contrasts with other state-of-the-art transductive few-shot learning methods (Qi et al., 2021; Lazarou et al., 2021; Zhu & Koniusz, 2023) that introduces additional time-consuming label propagation and GNNs. Note that previous work (Bendou et al., 2022; Qi et al., 2021; Zhu & Koniusz, 2023) employ an inference-time augmentation technique involving 30 times inference for each randomly cropped augmented sample, subsequently averaging the features for final prediction. For a fair comparison with the state-of-the-art methods (Bendou et al., 2022; Qi et al., 2021; Zhu & Koniusz, 2023), we adopt a similar approach for MetaFormer-A by shuffling the order of the samples 30 times and then computing the average of logits as the final prediction. We conduct the inference-time latency evaluation an NVIDIA RTX A6000 GPU for 5-way 5-shot scenarios. We extract features in advance for both protoLP (Zhu & Koniusz, 2023) and our MetaFormer and calculate the inference time without special parallel optimization. The results show that our MetaFormer-A with inference-time augmentation exhibits higher accuracies and remarkably superior computational efficiency, establishing a meaningful transductive baseline for pure transformer backbones in the realm of few-shot learning owing to its simplicity and efficiency.

Table 11: Comparison of average classification accuracy and inference times against state-of-the-art methods for 5-shot classification. $^\star$ means inference-time augmentation is used.

| Method | Backbone | Infer. Speed [ms] | *mini*ImageNet | *tiered*ImageNet | CIFAR-FS |
|---|---|---|---|---|---|
| CAN (Hou et al., 2019) | *ResNet-12* | - | $80.64_{\pm0.35}$ | $84.93_{\pm0.38}$ | − |
| EASY (Bendou et al., 2022) | *3 × ResNet-12* | - | $88.57_{\pm0.12}$ | $89.26_{\pm0.14}$ | $90.20_{\pm0.15}$ |
| ODC$^\star$ (Qi et al., 2021) | *WRN-28-10* | - | 88.22 | 91.20 | − |
| iLPC (Lazarou et al., 2021) | *WRN-28-10* | - | $88.82_{\pm0.42}$ | $92.46_{\pm0.42}$ | $90.60_{\pm0.48}$ |
| protoLP$^\star$ (Zhu & Koniusz, 2023) | *WRN-28-10* | 40.61 | $90.02_{\pm0.12}$ | $93.21_{\pm0.13}$ | $90.82_{\pm0.15}$ |
| MetaFormer-A$^\star$ | *ViT-Small* | **34.44** | $\mathbf{93.36_{\pm0.38}}$ | $\mathbf{93.66_{\pm0.50}}$ | $\mathbf{93.30_{\pm0.51}}$ |

## L  SETUP FOR IN-DOMAIN FEW-SHOT EVALUATION

### L.1  DATASETS USED FOR BENCHMARKS

For standard few-shot image classification evaluation with only class shift, we train and evaluate our MetaFormer presented in the main paper on the following few-shot benchmarks: *mini*ImageNet. (Vinyals et al., 2016b) is a subset of the ImageNet-1K, consisting of 100 classes and 600 images in each category. The classes are divided into 64, 16, and 20 for training, validation, and test, respectively.

*tiered*ImageNet. (Ren et al., 2018b) is another larger and more challenging subset of ImageNet-1K. It contains 34 higher-level nodes near the root of ImageNet, which are 608 classes in total. The dataset is split into 20, 6, and 8 higher-level nodes and corresponding 351, 97, and 160 classes as the training, validation, and testing set, respectively.

**CIFAR-FS** (Bertinetto et al., 2019) contains 100 classes and 600 images from the CIFAR100 dataset (Krizhevsky et al., 2009). The classes are split into 64 for training, 16 for validation, and 20 for testing.

**FC100** (Oreshkin et al., 2018) is built from the CIFAR100 (Krizhevsky et al., 2009) employing a splitting strategy analogous to that of the *tiered*ImageNet dataset to enhance difficulty, giving rise to 60 training, 20 validation, and 20 test classes.

### L.2  ADDITIONAL IMPLEMENTATION DETAILS

**Pretraining.** For MetaFormer, we adhere to the strategy delineated by  Hiller et al. (2022) for pretraining our vision transformer backbones on the meta-training split of each dataset, maintaining most of the training hyperparameter configurations reported in their study. Concretely, we employ

default two global crops and ten local crops with respective crop scales of (0.4, 1.0) and (0.05, 0.4). We use the image resolution of $224 \times 224$ and the output is projected to 8192 dimensions. A patch size of 16 and window size of 7 are used for aligning standard settings in ViT-small (Dosovitskiy et al., 2020; Touvron et al., 2021) and Swin-tiny (Liu et al., 2021), respectively. A batch size of 512 and a cosine-decaying learning rate schedule are used. For SMKD-MetaFormer, we follow Lin et al. (2023) to train the vision transformer backbones with the image resolution of $480 \times 480$.

**Meta-tuning.** We integrate our proposed SAM and TAM into the original vision transformer in every other layer, starting from the 6th layer, to construct holistic attention for meta-learning. Here, SAM and TAM are randomly initialized. The number of task probe vectors $T$ is configured to one for ViT and eight for Swin, and the pool size is set to $Z = 50$. Sampling size is set to $r = 15$ for the context class prototype update in the autoregressive setting. During meta-tuning, we follow most of the training techniques used in FewTURE (Hiller et al., 2022). We employ the SGD optimizer, utilizing a cosine-decaying learning rate initiated at $2 \times 10^{-4}$, a momentum value of 0.9, and a weight decay of $5 \times 10^{-4}$ across all datasets. The input image size is set to $224 \times 224$ for MetaFormer and $360 \times 360$ for SMKD-MetaFormer. Typically, training is conducted for a maximum of 200 epochs. To mitigate the risk of overfitting, we adopt the early stopping strategy coupled with freezing parameters of the first three layers. All additional hyperparameters are selected on 600 randomly sampled episodes from the respective validation sets to ascertain the optimal parameter configuration. For the evaluation of few-shot learning, we conduct a random sampling of 600 episodes from the test set to evaluate our model.

## M  SETUP FOR CROSS-DOMAIN FEW-SHOT EVALUATION

### M.1  DATASETS USED FOR BENCHMARKS

We use *mini*ImageNet as the source dataset for meta-training and perform the cross-domain few-shot evaluation on eight datasets with varying domain similarity, following Oh et al. (2022). The datasets can be separated into two groups: **BSCD-FSL benchmark** (Guo et al., 2020) and **nonBSCD-FSL**. For BSCD-FSL benchmark (CropDisease, EuroSAT, ISIC, ChestX), we follow Guo et al. (2020) for the dataset split. And for nonBSCD-FSL benchmark (CUB, Car, Plantaem Places), we follow Tseng et al. (2020) for the splitting procedure. We refer to Oh et al. (2022) for a more detailed description of each dataset.

### M.2  IMPLEMENTATION DETAILS

For cross-domain experiments, we meta-train our MetaFormer-I and MetaFormer-A on the *mini*ImageNet dataset as in Section 4.1 in the main paper and then freeze all parameters during evaluation on cross-domain benchmarks. We follow the standard meta-test procedure to calculate the performance of baseline models (Hiller et al., 2022; Lin et al., 2023) trained on *mini*ImageNet.

### M.3  RESULTS.

## N  SETUP FOR MULTI-DOMAIN FEW-SHOT EVALUATION

### N.1  DATASETS USED FOR BENCHMARKS

**Meta-Dataset** (Triantafillou et al., 2020) is a more challenging and realistic large-scale benchmark consisting of ten image datasets including ImageNet-1k, Omniglot, Aircraft, CUB, Textures, Quick-Draw, Fungi, VGG Flower, Traffic Signs, and MSCOCO, each with specified train, val and test splits. We follow Hu et al. (2022) to utilize the train and val splits of the initial eight datasets (in-domain) for meta-training and validation, while employing the test splits of all datasets for meta-testing. We refer to Triantafillou et al. (2020) for an in-depth exploration of Meta-Dataset.

### N.2  IMPLEMENTATION DETAILS

We meta-train both PMF (Hu et al., 2022) and our MetaFormer build upon the same pre-trained vision transformer (Caron et al., 2021) in a 5-way 1-shot setting, adhering to most of the unchanged training

Table 12: Broader study of cross-domain few-shot learning. Average classification accuracy (%) for 5-way 1-shot and 5-way 5-shot scenarios when meta-learning on *mini*ImageNet (Vinyals et al., 2016a) but meta-testing on cross-domain few-shot benchmarks similar and dissimilar to *mini*ImageNet. Reported are the mean and 95% confidence interval.

(a) Cross-domain few-shot benchmarks similar to *mini*ImageNet.

| Setting | Method | BackBone | CUB | Cars | Places | Plantae |
|---|---|---|---|---|---|---|
| 1-shot | FewTURE (Hiller et al., 2022) | *ViT-Small* | $48.21_{\pm 0.83}$ | $33.97_{\pm 0.63}$ | $58.74_{\pm 0.91}$ | $43.31_{\pm 0.76}$ |
| | MetaFormer-I (Ours) | *ViT-Small* | $58.23_{\pm 0.84}$ | $40.13_{\pm 0.63}$ | $62.85_{\pm 0.85}$ | $51.58_{\pm 0.78}$ |
| | MetaFormer-A (Ours) | *ViT-Small* | $58.72_{\pm 0.86}$ | $38.48_{\pm 0.65}$ | $65.60_{\pm 0.92}$ | $52.20_{\pm 0.84}$ |
| | SMKD (Lin et al., 2023) | *ViT-Small* | $54.64_{\pm 0.84}$ | $34.30_{\pm 0.64}$ | $62.75_{\pm 0.92}$ | $45.57_{\pm 0.81}$ |
| | SMKD+MetaFormer-I (Ours) | *ViT-Small* | $58.56_{\pm 0.82}$ | $37.66_{\pm 0.63}$ | $62.90_{\pm 0.88}$ | $47.82_{\pm 0.79}$ |
| | SMKD+MetaFormer-A (Ours) | *ViT-Small* | $64.02_{\pm 0.89}$ | $38.37_{\pm 0.70}$ | $68.69_{\pm 0.97}$ | $49.06_{\pm 0.90}$ |
| 5-shot | FewTURE (Hiller et al., 2022) | *ViT-Small* | $67.70_{\pm 0.77}$ | $46.54_{\pm 0.73}$ | $74.70_{\pm 0.69}$ | $61.72_{\pm 0.71}$ |
| | MetaFormer-I (Ours) | *ViT-Small* | $77.02_{\pm 0.74}$ | $53.17_{\pm 0.66}$ | $80.92_{\pm 0.64}$ | $68.61_{\pm 0.71}$ |
| | MetaFormer-A (Ours) | *ViT-Small* | $75.58_{\pm 0.82}$ | $52.42_{\pm 0.74}$ | $82.65_{\pm 0.66}$ | $67.59_{\pm 0.80}$ |
| | SMKD (Lin et al., 2023) | *ViT-Small* | $77.17_{\pm 0.69}$ | $50.72_{\pm 0.71}$ | $80.79_{\pm 0.63}$ | $64.90_{\pm 0.72}$ |
| | SMKD+MetaFormer-I (Ours) | *ViT-Small* | $79.83_{\pm 0.67}$ | $56.09_{\pm 0.63}$ | $83.00_{\pm 0.61}$ | $68.53_{\pm 0.70}$ |
| | SMKD+MetaFormer-A (Ours) | *ViT-Small* | $81.36_{\pm 0.66}$ | $55.65_{\pm 0.62}$ | $84.19_{\pm 0.62}$ | $69.49_{\pm 0.70}$ |

(b) Cross-domain few-shot benchmarks dissimilar to *mini*ImageNet.

| Setting | Method | BackBone | CropDisease | EuroSAT | ISIC | ChestX |
|---|---|---|---|---|---|---|
| 1-shot | FewTURE (Hiller et al., 2022) | *ViT-Small* | $68.22_{\pm 0.88}$ | $61.77_{\pm 0.81}$ | $28.67_{\pm 0.56}$ | $22.60_{\pm 0.44}$ |
| | MetaFormer-I (Ours) | *ViT-Small* | $74.16_{\pm 0.83}$ | $67.73_{\pm 0.76}$ | $36.96_{\pm 0.57}$ | $27.06_{\pm 0.43}$ |
| | MetaFormer-A (Ours) | *ViT-Small* | $78.93_{\pm 0.81}$ | $69.70_{\pm 0.81}$ | $36.10_{\pm 0.60}$ | $27.37_{\pm 0.40}$ |
| | SMKD (Lin et al., 2023) | *ViT-Small* | $75.99_{\pm 0.82}$ | $69.36_{\pm 0.81}$ | $34.00_{\pm 0.63}$ | $22.59_{\pm 0.41}$ |
| | SMKD+MetaFormer-I (Ours) | *ViT-Small* | $76.01_{\pm 0.82}$ | $70.53_{\pm 0.77}$ | $37.52_{\pm 0.60}$ | $26.54_{\pm 0.41}$ |
| | SMKD+MetaFormer-A (Ours) | *ViT-Small* | $83.11_{\pm 0.77}$ | $76.14_{\pm 0.82}$ | $38.38_{\pm 0.66}$ | $26.22_{\pm 0.43}$ |
| 5-shot | FewTURE (Hiller et al., 2022) | *ViT-Small* | $86.41_{\pm 0.56}$ | $77.88_{\pm 0.57}$ | $38.53_{\pm 0.54}$ | $25.54_{\pm 0.43}$ |
| | MetaFormer-I (Ours) | *ViT-Small* | $88.52_{\pm 0.60}$ | $85.73_{\pm 0.50}$ | $52.32_{\pm 0.57}$ | $35.82_{\pm 0.54}$ |
| | MetaFormer-A (Ours) | *ViT-Small* | $87.15_{\pm 0.69}$ | $86.42_{\pm 0.52}$ | $51.28_{\pm 0.62}$ | $35.32_{\pm 0.50}$ |
| | SMKD (Lin et al., 2023) | *ViT-Small* | $92.11_{\pm 0.45}$ | $85.99_{\pm 0.52}$ | $47.58_{\pm 0.62}$ | $26.28_{\pm 0.42}$ |
| | SMKD+MetaFormer-I (Ours) | *ViT-Small* | $89.69_{\pm 0.52}$ | $85.08_{\pm 0.56}$ | $54.32_{\pm 0.61}$ | $35.68_{\pm 0.50}$ |
| | SMKD+MetaFormer-A (Ours) | *ViT-Small* | $91.17_{\pm 0.50}$ | $87.12_{\pm 0.54}$ | $54.59_{\pm 0.61}$ | $35.66_{\pm 0.48}$ |

hyperparameters reported in PMF, with minimal alterations. The training process spans 100 epochs, utilizing the SGD optimizer with a momentum of 0.9. A cosine-decaying learning rate scheduler is employed, initialized at $5e - 4$.

N.3   RESULTS.

We evaluate the effectiveness of MetaFormer on the large-scale and challenging Meta-Dataset. Table 13 presents the test accuracy measured on each dataset meta-test set. MetaFormer outperforms previous meta-learning SOTA method PMF (Hu et al., 2022) in multi-domain adaptation scenarios. Its superior performance, especially in settings with scarce samples (e.g., one sample per category), underscores the efficacy of our proposed approach for fast adaptation in each domain.

Table 13: Broader study of multi-domain few-shot learning. Average classification accuracy (%) for 5-way 1-shot and variable-way variable-shot scenarios.

| Model | In-domain | | | | | | | | Out-of-domain | | Avg |
|---|---|---|---|---|---|---|---|---|---|---|---|
| | INet | Omglot | Acraft | CUB | DTD | QDraw | Fungi | Flower | Sign | COCO | |
| *5-way 1-shot* | | | | | | | | | | | |
| PMF (Hu et al., 2022) | 56.35 | 94.22 | 88.00 | 84.63 | **52.90** | 75.18 | 84.13 | 75.20 | 55.02 | 49.69 | 71.53 |
| MetaFormer-I (Ours) | **63.41** | **94.57** | 87.93 | **89.17** | 51.33 | 75.10 | 81.97 | **85.06** | **57.33** | **53.64** | **73.95** |
| MetaFormer-A (Ours) | 66.03 | 96.47 | 89.75 | 91.95 | 52.20 | 79.17 | 84.44 | 88.88 | 58.89 | 58.12 | 76.59 |
| *variable-way variable-shot* | | | | | | | | | | | |
| PMF (Hu et al., 2022) | 74.59 | 91.79 | 88.33 | 91.02 | **86.61** | 79.23 | 74.20 | 94.12 | 88.85 | **62.59** | 83.13 |
| MetaFormer-I (Ours) | **75.65** | **92.26** | **90.89** | **91.01** | 85.21 | **79.76** | **75.73** | **97.13** | **90.29** | 60.25 | **83.82** |

