# OpenReview forum: "MetaFormer with Holistic Attention Modelling Improves Few-Shot Classification"
_ICLR.cc/2024/Conference — ICLR 2024 Conference Withdrawn Submission_

### Official Review · Reviewer_xXZf · 2023-10-25

**Soundness:** 3 good
**Presentation:** 3 good
**Contribution:** 2 fair
**Rating:** 5
**Confidence:** 4

**Summary:**

A coherent and lightweight framework MetaFormer is proposed to improve the Vision Transformer performance in meta-learning. The framework contains a sample-level attention module (SAM) and a task-level attention module (TAM). The SAM enables the consistency of attended features across samples in a task whilst the TAM regularizes the learning of features for the current task by attending to a specific task in the pool. Extensive experiments are conducted on four commonly used FSL benchmark datasets and the new SOTA is achieved with a marginal increase in computational cost.

**Strengths:**

-- The paper adds inter-sample and inter-task attention modules to the original Vision Transformer for meta-learning. These ideas are not novel whilst the implementation of them in such a way is somewhat novel.

-- The presentation is generally good but lacks clarity for some details (see weaknesses).

-- Thorough experiments on the in-domain and cross-domain settings provide valuable references to the community of FSL.

**Weaknesses:**

-- The authors use the terms context and target sets instead of support and query sets which are usually employed in FSL literature. This makes the presentation harder to understand.

-- The main concern is the use of test samples during learning which makes it a transductive learning method. It might be that I misunderstand the training details of the model but it is unclear to me if M unlabelled query samples are used in any way before final prediction using Eq(7).

-- The autoregressive inference setting employs the information from the test data. It is unfair to compare with those under the true inductive learning setting.

-- It is unclear what properties of the learned task embedding have. A suggestion would be to analyze the task embedding space somehow to give intuitive insights for better understanding. For example, one can sample N tasks from the meta-testing data and M tasks from the meta-training data and compute the NxM distance matrix and take a closer look at some exemplar rows/columns to see what kind of tasks are similar in the learned task embedding space.

**Questions:**

1. In tables 1 and 2,  why are the bottom two groups of methods separated whilst they use the same backbone ViT-Small?

2. If I misunderstand the method regarding the transductive setting, is it possible to adapt the framework to the transductive setting and how?

---

> ### Author Response · Authors · 2023-11-21
> **Response to Reviewer xXZf (Part 1)**
>
> We extend our appreciation for your  constructive feedback on our manuscript. Below, we address each of your points comprehensively. Should there be any additional queries or clarifications needed, please feel free to let us know.
>
> #### Q1: The terms context and target sets instead of support and query sets
> > Thanks for your suggestion. We have revised our manuscript to use "support" and "query" sets to align with more FSL works and enhance clarity. Our initial choice of "context" and "target" aligns with previous meta-learning works of [1-4],  where "context" set provide conditions for task adaptation and "target" set serves evaluation targets of the adapted model.
> >
> > [1] Zintgraf, et al. “Fast Context Adaptation via Meta-Learning.” In ICML, 2019
> >
> > [2] Rusu, et al. “Meta-learning with latent embedding optimization.” In ICLR, 2019
> >
> > [3] Requeima, et al. “Fast and ﬂexible multi-task classiﬁcation using conditional neural adaptive processes.” In NeurIPS, 2019.
> >
> > [4] Patacchiola, et al. “Contextual Squeeze-and-Excitation for Efficient Few-Shot Image Classification.” In NeurIPS, 2022
>
> #### Q2: Whether the proposed method is inductive or transductive
> > - We would like to humbly clarify that the proposed MetaFormer with its meticulously designed sample causal mask **flexibly and efficiently accommodates both inductive and transductive settings**. This also constitutes one of our **core contributions**.
> >    - When configuring the sample causal mask as the inductive variant (refer to Figure 2(a) for details), we instantiate MetaFormer-I. This design choice with **query interactions blocked**, coupled with the inherent Layer Normalization in vision transformers, ensures **independent predictions for each query sample**, adhering strictly to inductive learning principles.
> >    - When configuring the sample causal mask as the autoregressive variant (refer to Figure 2(b) for details), we instantiate **MetaFormer-A**. This represents a **coherent and efficient transductive implementation** capable of extracting support and query feature embeddings **in a single feedforward pass**. To the best of our knowledge, MetaFormer-A stands out as the **first pure transformer-backed method for transductive few-shot image classification**.  It eliminates the need for extra time-consuming components suh as label propagation in GNNs [1] or specialized loss functions [2], offering simplicity and efficiency.
> > - We sincerely appreciate the reviewer's great suggestion to include more comparisons validating the superiority of our transductive version, MetaFormer-A. In response,
> >   - We conduct additional comparisons, pitting MetaFormer-A against other transductive methods. The results, in the table below, demonstrates its superior performance with a 79.41% accuracy without introducing extra components like label propagation in GNNs [1] or specialized loss functions [2], establishing it as a **compelling choice for seamless transitioning from the inductive setting**.
> >
> >     | Method        | Backbone |miniImageNet 5-way 1-shot |
> >     | ------------- | -------- |------------- |
> >     | TPN [1]       | ResNet   | 59.46        |
> >     | ADV [2]       | WRN      | 74.63        |
> >     | MetaFormer-A  | ViT-Small | **79.41**       |
> >
> >    - We further conduct a comparative analysis employing a naive transductive approach, wherein constraints on query interactions as MetaFormer-A impose are omitted, and the sampling size ($r$) is set to be equal to the number of query samples. This method is analogous to the one described in [3]. The following table results showcase the **superiority of our autoregressive version MetaFormer-A**.
> >
> >      | Method        | Backbone |miniImageNet 5-way 1-shot |
> >      | ------------- | -------- |------------- |
> >      | naive transductive | ViT-Small | 76.00    |
> >      | MetaFormer-I  | ViT-Small | 75.78       |
> >      | MetaFormer-A  | ViT-Small | **79.41**       |
> >
> >
> > [1] Liu, et al. "Learning to propagate labels: Transductive propagation network for few-shot learning." arXiv preprint arXiv:1805.10002, 2018.
> >
> > [2] Huang, et al. "Improving Task-Speciﬁc Generalization in Few-Shot Learning via Adaptive Vicinal Risk Minimization." In NeurIPS, 2022
> >
> > [3] Hou, et al. "Cross attention network for few-shot classification." In NeurIPS, 2019
>
> #### Q3: Visualization analysis of learned task embeddings
>
> > Thank you for your insightful suggestion. We conduct a qualitative analysis of learned task probe vectors across different tasks randomly sampled from meta-train (M=3) and meta-test sets (N=3). As shown in Appendix J Fig. 7, the visualization effectively reveals the **efficacy of these vectors in capturing task relationships**. For example, we observe a higher similarity in task features among tasks involving car tires, dogs, and long-legged animals. This demonstrates MetaFormer's capability in discerning and utilizing task semantics.

---

> ### Author Response · Authors · 2023-11-21
> **Response to Reviewer xXZf (Part 2)**
>
> #### Q4. In tables 1 and 2, why are the bottom two groups of methods separated whilst they use the same backbone ViT-Small?
>
> > We  apologize for any potential misunderstandings. Our intention in distinguishing between these two groups of methods is to **highlight the effectiveness of MetaFormer when employed in conjunction with diverse training paradigms**.
> >  - FewTURE [1] represents a pioneering meta-learning approach specifically tailored for self-supervised pre-trained Vision Transformers (ViT).
> >  - The SMKD [2] method adopts a transfer learning approach based on self-distillation.
> >
> > The results obtained from our experiments consistently showcase MetaFormer's ability to enhance task adaptation across both of these distinct methodological groups.
> >
> > [1] Hiller, et al. "Rethinking generalization in few-shot classification." In NeurIPS, 2022.
> >
> > [2] Lin, et al. "Supervised masked knowledge distillation for few-shot transformers." In CVPR, 2023

---

> ### Author Response · Authors · 2023-11-22
> **We would love to hear back from Reviewer xXZf**
>
> Hi Reviewer xXZf,
>
> We would like to follow up to see if our response addresses your concerns or if you have any further questions. We would really appreciate the opportunity to discuss this further if our response has not already addressed your concerns.
>
> Thank you again!

---

> > ### Comment · Reviewer_xXZf · 2023-11-22
> >
> > Thanks for the reply, much appreciated. The transductive learning method does not seem to perform comparably well with SOTA though the more advanced ViT backbone is employed.

---

> > > ### Author Response · Authors · 2023-11-23
> > > **Q: Gratitude to Reviewer xXZf's prompt response and updated fair comparison in the transductive setting**
> > >
> > > > We appreciate the reviewer’s prompt response very much, letting us know your remaining concern on the comparative performance of our autoregressive method, MetaFormer-A, with the current state-of-the-art transductive method.
> > > >
> > > >
> > > > We again highlight our core contribution that seamlessly transits between the inductive and transductive settings via simply configuring the sample causal mask.
> > > >   - Under the inductive setting, we have already demonstrated the unquestionable superiority of MetaFormer-I, agreed by the reviewer, over other state-of-the-art baselines.
> > > >   - Under the transductive setting, the proposed MetaFormer-A by only setting the autoregressive sample causal mask (see Fig 2(b)) requires **only one singe-pass during inference**, which contrasts with other state-of-the-art transductive few-shot learning methods [3-5] that introduces additional time-consuming label propagation and GNNs.
> > > >   - We conduct additional comparisons for MetaFormer-A against other SOTA transductive methods. Note that for a fair comparison with the state-of-the-art methods [2,3] that employ inference-time augmentation, we adopt a similar approach for MetaFormer-A. This involves shuffling the order of the samples in a meta-testing task 30 times and then computing the average of logits as the final prediction. We conduct the inference-time latency evaluation on an NVIDIA RTX A6000 GPU for 5-way 5-shot scenarios. We extract features in advance for both protoLP[5] and our MetaFormer, and calculate the inference time without special parallel optimization.
> > > >      - **Superiority over SOTA**: The results, reported below and in Appendix K Table 11, demonstrate that our MetaFormer-A with inference-time augmentation **exhibits not only higher accuracies but also remarkably superior computational efficiency.**
> > > >
> > > >
> > > > In conclusion, we believe that our MetaFormer-A establishes a meaningful transductive baseline for pure transformer backbones in the realm of few-shot learning owing to its simplicity and efficiency. We have incorporated this part of experiments in our revision.
> > > >
> > > > | Method | Backbone | Inference Speed[ms] | miniImageNet 5-shot | tieredImageNet 5-shot | CIFAR-FS 5-shot |
> > > > | --- | --- | --- | --- | --- | --- |
> > > > | CAN [1] | ResNet-12 | - | 80.64 ± 0.35 | 84.93 ± 0.38 | - |
> > > > | EASY [2] | 3*ResNet-12 | - | 88.57 ± 0.12 | 89.26 ± 0.14 | 90.20±0.15 |
> > > > | ODC [3] | WRN-28-10 | - | 88.22 | 91.20 | - |
> > > > | iLPC [4] | WRN-28-10 | - | 88.82 ± 0.42 | 92.46 ± 0.42 | 90.60±0.48 |
> > > > | protoLP [5] | WRN-28-10 | 40.61 | 90.02 ± 0.12 | 93.21 ± 0.13 | 90.82±0.15 |
> > > > | MetaFormer-A | ViT-Small | 34.44 | 93.36±0.38 | 93.66 ± 0.50 | 93.30 ± 0.51 |
> > > >
> > > > [1] Hou, et al. "Cross attention network for few-shot classification." In NeurIPS, 2019
> > > >
> > > > [2] Bendou et al. "Easy—ensemble augmented-shot-y-shaped learning: State-of-the-art few-shot classification with simple components." In Journal of Imaging, 2022
> > > >
> > > > [3] Qi et al. "Transductive few-shot classification on the oblique manifold." In ICCV, 2021
> > > >
> > > > [4] Lazarou et al., “Iterative label cleaning for transductive and semi-supervised few-shot” learning. In ICCV, 2021
> > > >
> > > > [5] Zhu, et al., “Transductive Few-shot Learning with Prototype-based Label Propagation by Iterative Graph Refinement” In CVPR, 2023

---

### Official Review · Reviewer_xYfp · 2023-10-28

**Soundness:** 3 good
**Presentation:** 3 good
**Contribution:** 2 fair
**Rating:** 5
**Confidence:** 3

**Summary:**

This paper proposed MetaFormer, a new Transformer-based method for meta-learning. To improve the efficiency of self-attention in meta-learning, the authors propose to distengle the computation into three different dimensions: Task Attention, Spatial Attention and Sample Attention. Compared to existing state-of-the-arts, the proposed method achieves better performance across different datasets.

**Strengths:**

1.  The motivation of this paper is clear: most exisiting meta-learning frameworks only show effectiveness in convolutional neural networks, as Transformer is prevailing these days, it is meaningful to validate and adapt this architecture in meta-learning as well.
2. The proposed method achieves clearly better performance than SOTA methods.
3. This paper is easy to follow. The figures well illustrate the framework of the proposed MetaFormer.

**Weaknesses:**

1. Recent large-scale pretrained vision foundation models, such as CLIP and SAM, have demonstrated superior zero-shot performance on image classification and visual grounding. In this context, one of my primary concerns is that the problem setting in this paper is not sufficiently significant. For example, datasets such as miniImageNet and CIFAR-FS may provide insights into the performance of meta-learning frameworks on small datasets, but they cannot accurately reflect performance on large-scale open-vocabulary datasets.

2. From a technical perspective, the proposed holistic attention mechanism is novel in meta-learning. However, in general, the decoupling of self-attention computation into multiple dimensions is not new in the literature [1, 2]. For example, in video processing, TimeSformer [2]  has proposed to separate the spatial and temporal attention in a single block.

3. The paper does not demonstrate any efficiency gains from the proposed attention design. This is unconvincing, as one of the primary motivations described in the introduction is to reduce the computational cost of attention in ViTs when adapting for Meta-learning.

[1] Ho, Jonathan, et al. "Axial attention in multidimensional transformers." arXiv preprint arXiv:1912.12180 (2019).

[2] Bertasius, Gedas, Heng Wang, and Lorenzo Torresani. "Is space-time attention all you need for video understanding?." ICML. Vol. 2. No. 3. 2021.

**Questions:**

Can the authors report the FLOPs and inference speed, memory cost in Table 1, 2, 3? Or at least the Table 3.

---

> ### Author Response · Authors · 2023-11-21
> **Response to Reviewer xYfp (Part 1)**
>
> We appreciate very much your constructive comments on our paper. Please kindly find our response to your comments below, and all revisions made to the paper are highlighted in red for your ease of reference. We hope that our response satisfactorily addresses the issues you raised. Please feel free to let us know if you have any additional concerns or questions.
>
> #### Q1. Recent large-scale pretrained vision foundation models, such as CLIP and SAM, have demonstrated superior zero-shot performance. In this context, one of my primary concerns is that the problem setting in this paper is not sufficiently significant. Datasets such as miniImageNet and CIFAR-FS cannot accurately reflect performance on large-scale open-vocabulary datasets.
> > - Our paper is **expressly designed to enhance the few-shot performance of pre-trained vision foundation models** on downstream tasks, as underlined in the Introduction (last sentence in the first paragraph).
> >   - Pre-trained vision model: In our previous experiment, we adopt the **ViT pre-trained by DINO** as the pre-trained vision model for meta-training. Extensive empirical results robustly validate the effectiveness of our proposed method in enhancing few-shot learning performance on the pre-trained model.
> >   - Datasets: Despite the relatively modest scale of miniImageNet ad CIFAR-FS, our performance on these datasets **offers valuable insights into open-vocabulary datasets.** We follow the practice in FewTURE by pre-training DINO on only the meta-training set of the two datasets. This mirrors real-world scenarios where open-vocabulary downstream datasets are often out of the distribution of the pre-training dataset.
> > - **Existing literature highlights the potential for further enhancing CLIP's few-shot performance on downstream tasks utilizing few-shot techniques** [3-5]. Many of these innovative designs and key insights originate from meta-learning experiments conducted on small datasets. For instance, the cache model in [3] is akin to matching networks [6], the context conditional prompt in [4] is inspired by task-specific vectors [7], and the observation that refined channels focus more on foreground regions [5] correlates with findings in [8] about varying channel importance across tasks.
> > - To demonstrate the **compatibility of the proposed MetaFormer with larger pre-trained vision foundation models such as CLIP** and the effectiveness on open-vocabulary datasets, we adapt our method to the CLIP model with ViT-B/16 for advancing its few-shot performance in downstream tasks.
>     >   - Baselines: We compare with **(1)** zero-shot CLIP, **(2)**  TiP-Adapter [3], a state-of-the-art method that adopts the shot feature and its corresponding label for the key and value of the cache model as the classifier head, **(3)** the variant TiP-Adapter-F [3] that fine-tunes the key in the cache model, and **(4)** the variant TiP-Adapter-F with more layers fine-tuned, being comparable to ours in the number of parameters.
>     >   - Datasets: We follow [3] and evaluate on the challenging open-vocabulary datasets, EuroSAT [9] and ISIC [10].
>     >   - Training and evaluation protocol: We adopt the episodic approach as described in [6] to construct tasks: **(1)** during training, both support and query sets are sampled from the train set; **(2)** during evaluation,  we strictly follow the TiP-Adapter [3] pipeline for sampling the support set from the train set and the query set from the test set to construct a task for evaluation. This is reasonable as there are no new classes in the test set.
> >   - Implementation: Our implementation follows TiP-Adapter [3], integrating its cache model as the auxiliary classifier head. We exclusively fine-tune the SAM modules and the head while keeping the visual and textual encoders of CLIP frozen. Evaluation employs pre-trained word embeddings of a single prompt, “a photo of a [CLASS].” Further implementation details are available in Appendix I.
> >   - Results: As presented in the table below, highlight noteworthy insights: **(1)** CLIP pre-trained on large-scale web-crawled image-text pairs struggles with downstream datasets exhibiting a substantial domain gap, such as the medical dataset of ISIC; **(2)** adapting CLIP with a downstream dataset is pivotal to improved performance, with a caution against the risks of overfitting with excessive parameter adaptation; **(3)** our method significantly enhances Zero-shot CLIP on EuroSAT by 42.76% and ISIC by 43.81%, and its adaptation ability also **surpasses Tip-Adapter by a large margin**.
> >  - We sincerely thank the reviewer for the constructive comments. We have also incorporated these discussion into Appendix I.

---

> > ### Author Response · Authors · 2023-11-21
> > **Response to Reviewer xYfp (Part 1 Continued)**
> >
> > >
> > >| Method                        | EuroSAT       | ISIC          |
> > >| ----------------------------- | ------------- | ------------- |
> > >| Zero-shot CLIP                | $48.73 \pm 0.98$ | $21.07 \pm 0.76$ |
> > >| Tip-Adapter                   | $69.85 \pm 0.75$ | $28.70 \pm 0.97$ |
> > >| Tip-Adapter-F                 | $72.01 \pm 0.97$ | $32.27 \pm 1.11$ |
> > >| Tip-Adapter-F with more layers| $51.95 \pm 0.86$ | $16.17 \pm 0.78$ |
> > >| Tip-Adapter+MetaFormer (Ours) | $88.83 \pm 0.78$ | $45.96 \pm 1.44$ |
> > >
> > >[1] Hiller, et al. "Rethinking generalization in few-shot classification." In NeurIPS, 2022.
> > >
> > >[2] Hu, et al. "Pushing the limits of simple pipelines for few-shot learning: External data and fine-tuning make a difference". In CVPR, 2022
> > >
> > >[3] Zhang, et al. “Tip-adapter: Training-free clip-adapter for better vision-language modeling.” In ECCV, 2022
> > >
> > >[4] Zhou, et al. “Conditional prompt learning for vision-language models.” In CVPR, 2022
> > >
> > >[5] Zhu, et al. “Not All Features Matter: Enhancing Few-shot CLIP with Adaptive Prior Reﬁnement.” In ICCV, 2023
> > >
> > >[6] Snell, et al. “Matching networks for one shot learning". In NeurIPS, 2016
> > >
> > >[7] Oreshkin, et al. “TADAM: task dependent adaptive metric for improved few-shot learning.” In NeurIPS, 2018
> > >
> > >[8] Luo, et al. “Channel Importance Matters in Few-Shot Image Classiﬁcation.” In ICML, 2022
> > >
> > >[9] Helber, et al. "Eurosat: A novel dataset and deep learning benchmark for land use and land cover classification", In IEEE Journal of Selected Topics in Applied Earth Observations and Remote Sensing, 2019.
> > >
> > >[10] Tschandl, et al. "The HAM10000 dataset, a large collection of multi-source dermatoscopic images of common pigmented skin lesions", In Scientific data, 2018

---

> ### Author Response · Authors · 2023-11-21
> **Response to Reviewer xYfp (Part 2)**
>
> #### Q2. From a technical perspective, the proposed holistic attention mechanism is novel in meta-learning. However, in general, the decoupling of self-attention computation into multiple dimensions is not new in the literature [1, 2]
> > - We  would like to underscore the following novel and meaningful technical innovations made to the field of few-shot learning.
> >   - The decoupling of temporal and spatial attention has indeed been explored in video transformers [1, 2]. However, it is crucial to highlight that our consideration of the sample-to-sample relationship in few-shot learning presents a unique challenge distinct from the frame-to-frame relationship in videos, i.e., query samples have to be differentiated from support ones. Our introduction of **sample causal masks** serves an effective solution to address the challenge.
> >      - Notably, a mere adjustment in the design of these sample causal masks allows our method to **flexibly accommodate both inductive and autoregressive inference**.
> >      - We substantiate the effectiveness of the proposed masks in **Appendix D Table 5c**, concluding that a lack of effective constraints between support and query samples (see Appendix D Fig 6 for details of the ablated masks of within-support and support-query) results in sub-optimal performance.
> >   - The proposed task attention module (TAM) with **novel knowledge encoding and consolidation mechanisms** also contributes to few-shot learning within the context of vision transformers. Detailed insights into TAM's contributions are provided in response to reviewer kzi2 Q2.
> > - While we acknowledge that our proposed technique may not be characterized as groundbreaking in a broader sense, it is important to note its wide impact to the domain of few-shot learning. In particular:
> >   - The matching of local structural patterns between samples has long proven effective in few-shot learning [2, 3], though preceding endeavors relying on CNNs such as DeepEMD [2] are very slow. For example, DeepEMD demands 8 hours to evaluate 2000 5-way 5-shot episodes) [3]. We first **enable this pattern matching across both samples and tasks within the ideally fitting framework of ViT in a highly efficient manner** (requiring only 3.5 minutes). Notably, the success of ViT plays a pivotal role in explaining our markedly improved performance than prior CNN-based methods (miniImageNet 5-way 1-shot: Ours: 75.78% / DeepEMD: 65.91%).
> >   - Our decoupling of spatial and sample attention **makes the proposed method seamlessly compatible with recent state-of-the-art pre-trained vision transformers**, as mentioned in the Introduction, further enhancing their few-shot learning performances.
> >     - Our current experiments have already demonstrated the compatibility and effectiveness when working with the pre-trained model of **DINO**;
> >     - During the response period, we have also applied our approach to **CLIP**, achieving remarkable success by outperforming the SOTA method by 16.82%. Please find details to the response to Q1.
> >
> > In conclusion, we hope that these clarifications underscore the significance of our work in the field of meta-learning, and have cited/discussed these two works in the main text.
> >
> > [1] Ho, Jonathan, et al. "Axial attention in multidimensional transformers." arXiv preprint arXiv:1912.12180 (2019).
> >
> > [2] Bertasius, Gedas, Heng Wang, and Lorenzo Torresani. "Is space-time attention all you need for video understanding?." ICML. Vol. 2. No. 3. 2021.

---

> ### Author Response · Authors · 2023-11-21
> **Response to Reviewer xYfp (Part 3)**
>
> #### Q3 and Q4. Efficiency gains in FLOPs and inference speed, memory cost
>
> > We appreciate the reviewer's invaluable feedback, and delve into a meticulous comparative analysis of computational efficiency.
> > - This assessment encompasses (1) MetaFormer, (2) a naive implementation of sample-to-sample interactions without decoupling the spatial attention and sample attention, denoted as MetaFormer-naive, and (3) the selection of two state-of-the-art and representative methods with comparable parameter counts from the meta-learning and transfer-learning domains as our primary comparison objects (4) the additional computational ablation analysis in Table 3.
> >
> > - We conduct the evaluation on an NVIDIA RTX A6000 GPU, wherein we report performance metrics, including **inference-time GFLOPs, latency, and memory usage**, specifically for 5-way 1-shot and 5-way 5-shot scenarios on the miniImageNet dataset.
> > - The tables below distinctly signify that
> >   - MetaFormer significantly **reduces computational complexity in stark contrast to MetaFormer-naive**, the naive implementation of sample-to-sample interactions.
> >   - Compared to other baselines, MetaFormer exhibits not only **higher accuracies** attributable to comprehensive sample-to-sample interactions, but also **remarkably superior computational efficiency**.
> >   - Different combinations of these modules reveal **trade-offs between model complexity and performance**, as well as the benefits of each module in enhancing the model's capabilities.
> >
> > We have incorporated this detailed comparative analysis into Appendix E Table 6 in the revised manuscript, providing a robust understanding of the computational efficiency landscape and affirming the advantages of MetaFormer over baselines in efficiency.
> > | Method | GFLOPs | 5w1s Acc. | 5w1s Infer. GPU Memory | 5w1s Infer. Speed [ms] | 5w5s Acc. | 5w5s Infer. GPU Memory | 5w5s Infer. Speed [ms] |
> > | --- | --- | --- | --- | --- | --- | --- | --- |
> > | FewTURE  | 5.01 | $68.02 \pm 0.88$ | 3304M | $77.35 \pm 0.47$ | $84.51 \pm 0.53$ | 6482M | $111.22 \pm 1.27$ |
> > | SMKD-Prototype | 12.58 | $74.28 \pm 0.18$ | 4288M | $137.58 \pm 0.66$ | $88.82 \pm 0.09$ | 4666M | $171.37 \pm 0.78$ |
> > | MetaFormer-naive | 602.40 | N/A | 20.85G | $417.05 \pm 0.51$ | N/A | 30.86G | $659.94 \pm 1.05$ |
> > | MetaFormer-I | 4.88 | $75.78 \pm 0.71$ | 3661M | $67.65 \pm 0.78$ | $90.02 \pm 0.44$ | 5887M | $105.72 \pm 1.06$ |
> >
> > | SAM | TAM | Add. Params. | GFLOPs | 5w1s Acc. | 5w1s Infer. GPU Memory | 5w1s Infer. Speed [ms] | 5w5s Acc. |5w5s Infer. GPU Memory | 5w5s Infer. Speed [ms] |
> > | --- | --- | --- | --- | --- | --- |--- |--- | --- |--- |
> > | $\checkmark$ | $\checkmark$ | $+3.57 \mathrm{M}$ | 4.88 | $75.78 \pm 0.71$ | 3661M | $67.65 \pm 0.78$ | $90.02 \pm 0.44$ | 5887M | $105.72 \pm 1.06$ |
> > | $\checkmark$ | $\times$     | $+2.01 \mathrm{M}$ | 4.77 | $74.64 \pm 0.76$ | 3602M | $67.56 \pm 0.81$ | $87.53 \pm 0.47$ | 5878M | $103.93 \pm 0.84$ |
> > | $\times$ | $\checkmark$     | $+1.56 \mathrm{M}$ | 4.68 | $73.63 \pm 0.75$ | 3590M | $54.51 \pm 0.49$ | $87.76 \pm 0.52$ | 5868M | $88.07 \pm 0.79$ |

---

> ### Author Response · Authors · 2023-11-22
> **We would love to hear back from Reviewer xYfp**
>
> Hi Reviewer xYfp,
>
> We would like to follow up to see if our response addresses your concerns or if you have any further questions. We would really appreciate the opportunity to discuss this further if our response has not already addressed your concerns.
>
> Thank you again!

---

### Official Review · Reviewer_5apc · 2023-10-29

**Soundness:** 2 fair
**Presentation:** 3 good
**Contribution:** 3 good
**Rating:** 6
**Confidence:** 5

**Summary:**

In this paper, the authors propose a ViT-based few-shot learning framework namely MetaFormer. Starting from vanilla ViT, the authors first propose Sample-leel Attention Module to reduce the computation cost and better intra-task interaction, and then propose Task-level Attention Module to enhance inter-task interactions for better feature representation. MetaFormer achieves promising performance on various few-shot learning benchmarks.

**Strengths:**

1. The motivation of "decoupling space-level and sample-level attention" is intuitive. Since this design can effectively reduce the computation cost meanwhile reduce sufficient unrelated attention values to ensure the quality of output features.

2. The performance of MetaFormer is promising. And the visualization results demonstrate the effectiveness of proposed methods.

3. This paper is well written and easy to reproduce.

**Weaknesses:**

1. Some critical ablation studies are lacked. e.g., the nubmer of task probe vectors (why choosing 1 for ViTs) and the size of knowledge pool (which number is better). Besides, for Table 3, the authors could introduce more variants (e.g., vanilla ViT with more layers) to support that: the performance improvement is from SAM / TAM but not more parameters.

2. Though the authors claim that using proposed holistic attention mechanism can significantly reduce the computation complexity, the authors still need to provide essential FLOPS / latency statistic to support the merit. For example, reporting baseline method and MetaFormer under 5-way 5-shot setting.

3. More questions regrading the details of the paper, please see Question section for detail.

**Questions:**

1. In line 1 of page 5, the authors claim that the complexity of O((NK + M)^2 + L^2). Nevertheless, both NK+M and L cannot be omitted in the decoupled attention, therefore it mighte be O(L(NK+M)^2 + (NK+M)L^2). The authors could recheck the complexity and ensure the correctness of the manuscript.

2. As shown in Eqn. 5, the tokens in knowledge pool are updated by direct addition without averaging. The authors could discuss the performance between with and without averaging during pool consolidation.

---

> ### Author Response · Authors · 2023-11-21
> **Response to Reviewer 5apc (Part 1)**
>
> Thank you sincerely for your thoughtful feedback on our work. Below, we have provided a detailed explanation for your concerns as follows. Please do not hesitate to let us know if you have any further questions.
>
> #### Q1. Ablation studies to support that: the performance improvement is from SAM / TAM but not more parameters
> > We have thoroughly examined the factors contributing to our performance improvement, concluding that the improvement is attributed to SAM/TAM rather than a mere expansion of parameters. This assessment is articulated through two primary perspectives.
> > - Comparison with other baselines with equivalent or more parameters: In the updated Tables 1 and 2, we observe that FewTURE/HCTransformers, despite possessing a larger number of parameters, markedly lags behind the proposed MetaFormer.
> >
> > - Ablation studies
> >   - We conduct the analysis by comparing with an ablated version, achieved by **naively augmenting the number of layers in ViT-Small to make it comparable with the proposed MetaFormer**. The results presented in the following table and in Appendix F Table 7 substantiate that merely increasing parameters cannot fully address the challenges inherent in few-shot learning. In fact, such augmentation may even elevate the risk of overfitting.
> >
> >     | Method                           | Backbone             | Backbone Params  | Total Params    | 1-shot           | 5-shot            |
> >     | -------------------------------- | -------------------- | ---------------- | --------------- | ---------------- | ----------------- |
> >     | vanilla ViT with more layers     | ViT-Small            | $21\mathrm{M}$   | $25.2 \mathrm{M}$ | $69.75 \pm 0.71$ | $84.12 \pm 0.56$  |
> >     | MetaFormer-I (Ours)              | ViT-Small            | $21\mathrm{M}$   | $24.5 \mathrm{M}$ | $75.78 \pm 0.71$ | $90.02 \pm 0.44$  |
> >     | MetaFormer-A (Ours)              | ViT-Small            | $21\mathrm{M}$   | $24.5 \mathrm{M}$ | $79.41 \pm 0.73$ | $91.21 \pm 0.44$  |
> >   - In **Table 3**, we present an ablation study where we remove either TAM or SAM, both of which yield a noticable performance drop. This underscores the indispensable contributions of both TAM and SAM to the overall effectiveness of our model.
> >   - In **Appendix D Table 5c**, We provide additional evidence of the effectiveness of the proposed SAM, by maintaining the number of parameters and only varying the masks.  The ablated masks of within-support and support-query (see Appendix D Fig 6 for details) manifest sub-optimal performance, further validating that SAM with the inductive mask works not because of the introduction of extra parameters.
> >   - We have also followed the reviewer's suggestion by investigating the impact of increasing the number of task probe vectors and pool size.  The results, as illustrated in the tables below, indicate that (1) our model is **not sensitive to the number of task probe vectors**, and (2) **a sufficiently diverse but compact knowledge pool** (rather than the largest pool) leads to improvements in performance.
> >
> >|  The number of probe vectors | miniImageNet 5-way 1-shot |
> >    | ------------- | ------------------------- |
> >    | 1  |    $75.78 \pm 0.71$       |
> >    | 4  |    $75.28 \pm 0.73$       |
> >    | 8  |    $75.44 \pm 0.72$       |
> >    | 16  |   $75.58 \pm 0.72$       |
> >
> >    | The size of knowledge pool |tieredImageNet 5-way 5-shot |
> >    | -------------  |------------------------- |
> >    | 10  | $90.40 \pm 0.49$       |
> >    | 50  | $91.44 \pm 0.53$       |
> >    | 100 | $89.92 \pm 0.54$       |

---

> ### Author Response · Authors · 2023-11-21
> **Response to Reviewer 5apc (Part 2)**
>
> #### Q2. Essential FLOPS / latency statistic to support the merit of reducing computational complexity. For example, reporting baseline method and MetaFormer under 5-way 5-shot setting.
> > We appreciate the reviewer's invaluable feedback, and delve into a meticulous comparative analysis of computational efficiency.
> > - This assessment encompasses (1) MetaFormer, (2) a naive implementation of sample-to-sample interactions without decoupling the spatial attention and sample attention, denoted as MetaFormer-naive, and (3) two other state-of-the-art methods.
> > - We conduct the evaluation on an NVIDIA RTX A6000 GPU, wherein we report performance metrics, including **inference-time GFLOPs and latency**, specifically for 5-way 1-shot and 5-way 5-shot scenarios on the miniImageNet dataset.
> > - The table below distinctly signifies that
> >   - MetaFormer significantly **reduces computational complexity in stark contrast to MetaFormer-naive**, the naive implementation of sample-to-sample interactions.
> >   - Compared to other baselines, MetaFormer exhibits  not only **higher accuracies** attributable to comprehensive sample-to-sample interactions, but also **remarkably superior computational efficiency**.
> >
> > We have incorporated this detailed comparative analysis into Appendix E Table 6 in the revised manuscript, providing a robust understanding of the computational efficiency landscape and affirming the advantages of MetaFormer over baselines in efficiency.
> >
> >| Method | GFLOPs | 5-way 1-shot Acc. | 5-way 1-shot Infer. Speed [ms] | 5-way 5-shot Acc. | 5-way 5-shot Infer. Speed [ms] |
> >| --- | --- | --- | --- | --- | --- |
> >| FewTURE | 5.01 | $68.02 \pm 0.88$ | $77.35 \pm 0.47$ | $84.51 \pm 0.53$ | $111.22 \pm 1.27$ |
> >| SMKD-Prototype | 12.58 | $74.28 \pm 0.18$ | $137.58 \pm 0.66$ | $88.82 \pm 0.09$ | $171.37 \pm 0.78$ |
> >| MetaFormer-naive | 602.40 | N/A | $417.05 \pm 0.51$ | N/A | $659.94 \pm 1.05$ |
> >| MetaFormer-I | 4.88 | $75.78 \pm 0.71$ | $67.65 \pm 0.78$ | $90.02 \pm 0.44$ | $105.72 \pm 1.06$ |
>
> #### Q3. The complexity should be O(L(NK+M)^2 + (NK+M)L^2).
> > We appreciate the reviewer for pointing out this typo. We have revisited and corrected the complexity calculation in the manuscript based on your suggestion.
>
> #### Q4. The performance between with and without averaging during pool consolidation in Eqn. 5
> > We conducted supplementary experiments to explore the impact of "averaging". As illustrated in the table below, the **performance difference between consolidation with averaging and without averaging is relatively marginal**. This can be elucidated by the nature of the cosine similarity-based score function employed during knowledge retrieval, which remains not influenced by the magnitude of the stored vectors in the pool. To this end, we adopt the addition without averaging.
> >
> >| Method          | miniImageNet 5-way 1-shot |
> >| --------------- | ------------------------- |
> >| with averaging    |        $75.78 \pm 0.71$   |
> >| without averaging |        $75.57 \pm 0.72$   |

---

> ### Author Response · Authors · 2023-11-22
> **We would love to hear back from Reviewer 5apc**
>
> Hi Reviewer 5apc,
>
> We would like to follow up to see if our response addresses your concerns or if you have any further questions. We would really appreciate the opportunity to discuss this further if our response has not already addressed your concerns.
>
> Thank you again!

---

### Official Review · Reviewer_kzi2 · 2023-10-31

**Soundness:** 3 good
**Presentation:** 3 good
**Contribution:** 2 fair
**Rating:** 5
**Confidence:** 3

**Summary:**

The paper introduces MetaFormer, a ViT-based framework, designed to excel in the domain of few-shot image classification. It splits attention mechanisms into two key phases: intra-task and inter-task interactions. Intra-task interactions are handled by the Sample-level Attention Module (SAM), which models sample relationships within tasks. For inter-task interactions, the Task-level Attention Module (TAM) is introduced to learn task-specific probe vectors and retrieve relevant semantic features from previous tasks, building a dynamic knowledge pool. MetaFormer demonstrates good performance across a wide range of benchmarks, including those related to few-shot learning and cross-domain tasks.

**Strengths:**

1)  The concept behind the proposed Metaformer is very simple and straightforward.
2)  The proposed Metaformer delivers superior quantitative results on extensive few-shot learning benchmarks.

**Weaknesses:**

1) Fundamentally, the technical contributions concerning sample-level attention and task-level attention presented in this work are not groundbreaking. For instance, the approach of decoupling attention (as detailed in section 3.2) to alleviate computational complexity is a well-established practice, particularly within the domain of video transformers.
2) A more thorough examination of related research is warranted. For example, it would be insightful to delve into the distinctions between the inter-task attention module in [1] and the proposed TAM module, even if [1] is rooted in the continual learning community.
3) It would enhance the clarity of Tables 1 and 2 to incorporate columns displaying the number of parameters for each backbone model, as opposed to segregating this information in the ablation study section. Such an adjustment would facilitate a more straightforward assessment of whether the observed improvements in numerical performance can be attributed to an augmented parameter count.

[1] Continual learning with lifelong vision transformer, CVPR 2022

**Questions:**

It would be beneficial to incorporate more in-depth discussions concerning prior research in the realm of meta-learning that incorporates vision transformers as their foundational architecture. For instance, when the authors highlight that "the majority of existing methods are specially tailored for CNNs and thus fail to translate effectively to vision transformers," it would be valuable to provide a more comprehensive explanation of the limitations of existing approaches when they are applied in conjunction with transformers.

---

> ### Author Response · Authors · 2023-11-21
> **Response to Reviewer kzi2 (Part 1)**
>
> We sincerely thank the reviewer for providing valuable feedback. We detail our response below point by point. Some experimental results have been updated in the revised paper, and any modifications made to the paper are highlighted in red for your convenience. Please kindly let us know whether you have any further concerns.
>
> #### Q1. Technical contributions concerning sample-level attention and task-level attention
>
> > - We acknowledge the reviewer's scrutiny of our technical contributions and would like to emphasize that our work indeed introduces novel and meaningful innovations to the field of few-shot learning.
> >   - The decoupling of temporal and spatial attention has indeed been explored in video transformers [1]. However, it is crucial to highlight that our consideration of the sample-to-sample relationship in few-shot learning presents a unique challenge distinct from the frame-to-frame relationship in videos, i.e., query samples have to be differentiated from support ones. Our introduction of **sample causal masks** serves an effective solution to address the challenge.
> >      - Notably, a mere adjustment in the design of these sample causal masks allows our method to **flexibly accommodate both inductive and autoregressive inference**.
> >      - We substantiate the effectiveness of the proposed masks in **Appendix D Table 5c**, concluding that a lack of effective constraints between support and query samples (see Appendix D Fig 6 for details of the ablated masks of within-support and support-query) results in sub-optimal performance.
> >   - The proposed task attention module (TAM) with **novel knowledge encoding and consolidation mechanisms** also contributes to few-shot learning within the context of vision transformers. Detailed insights into TAM's contributions are provided in response to Q2.
> > - While we acknowledge that our proposed technique may not be characterized as groundbreaking in a broader sense, it is important to note its wide impact to the domain of few-shot learning. In particular:
> >   - The matching of local structural patterns between samples has long proven effective in few-shot learning [2, 3], though preceding endeavors relying on CNNs such as DeepEMD [2] are very slow. For example, DeepEMD demands 8 hours to evaluate 2000 5-way 5-shot episodes) [3]. We first **enable this pattern matching across both samples and tasks within the ideally fitting framework of ViT in a highly efficient manner** (requiring only 3.5 minutes). Notably, the success of ViT plays a pivotal role in explaining our markedly improved performance than prior CNN-based methods (miniImageNet 5-way 1-shot: Ours: 75.78% / DeepEMD: 65.91%).
> >   - Our decoupling of spatial and sample attention **makes the proposed method seamlessly compatible with recent state-of-the-art pre-trained vision transformers**, as mentioned in the Introduction, further enhancing their few-shot learning performances.
> >     - Our current experiments have already demonstrated the compatibility and effectiveness when working with the pre-trained model of **DINO**;
> >     - During the response period, we have also applied our approach to **CLIP**, achieving remarkable success by outperforming the SOTA method by 16.82%. Please find details to the response to Q1 of Reviewer xYfp.
> >
> > In conclusion, we appreciate the reviewer's critical evaluation of our work, and we hope that these clarifications underscore the significance of our technical contributions to the field of few-shot learning.
> >
> >
> >[1] Bertasius, Gedas, Heng Wang, and Lorenzo Torresani. “Is space-time attention all you need for video understanding?.” ICML. Vol. 2. No. 3. 2021.
> >
> >[2] Zhang, Chi, Yujun Cai, Guosheng Lin, and Chunhua Shen. "Deepemd: Few-shot image classification with differentiable earth mover's distance and structured classifiers." CVPR, pp. 12203-12213. 2020.
> >
> >[3] Kang, Dahyun, Heeseung Kwon, Juhong Min, and Minsu Cho. "Relational embedding for few-shot classification." ICCV, pp. 8822-8833. 2021.

---

> ### Author Response · Authors · 2023-11-21
> **Response to Reviewer kzi2 (Part 2)**
>
> #### Q2. Distinctions between the inter-task attention module in [1] and the proposed TAM module, even if [1] is rooted in the continual learning community.
>
> > We appreciate the reviewer's great suggestion, and have incorporated the following discussions into the revised manuscript.
> >
> > - The TAM module proposed in our study distinguishes itself from the inter-task attention (IT-att) in [1] on several crucial fronts.
> >   - **Problem setting**: as acknowledged by the reviewer, TAM is rooted in the domain of *few-shot learning*, where the paramount concern is facilitating *knowledge transfer* between tasks. <u>In contrast</u>, IT-att in [1] is grounded in *continual learning*, where the primary focus lies in mitigating *catastrophic forgetting*.
> >   - **Task embedding**: while both TAM and IT-att [1] seemingly adopt a learnable embedding for each task, TAM utilizes it to represent *the knowledge specific to the current task*.  <u>In contrast</u>, IT-att stores *all past knowledge* in it through regularization-based consolidation mentioned below.
> >   - **Encoding of knowledge from other tasks**: Owing to disparate problem settings, TAM maintains *a knowledge pool that stores an array of task-dependent embeddings*. <u>In contrast</u>, IT-att [1] keeps a record of *a single key and a single bias*.
> >   - **Consolidation mechanism**: Leveraging our knowledge pool, we consolidate the current task probe vector by *averaging it with the most relevant vector in the pool* (refer to Eq.(5)). <u>In contrast</u>, IT-att, which is designed to address forgetting, employs *importance-based regularization* to enforce proximity of the current task embedding and previous one.
> > - Thus, the task interaction in TAM exhibits **greater flexibility and expressiveness**, aligning more closely with the objective of knowledge transfer in few-shot learning.
> >   - We further substantiate this claim through an ablation study, wherein we implement IT-att in our setting. The results, reported below and in Appendix G Table 8, demonstrate that our proposed **TAM consistently outperforms IT-att by approximately 1.3%** in the 5-way 5-shot setting on miniImageNet.
> >
> >      | Method             |   5-way 5-shot on miniImageNet |
> >      | ------------------ |  ------------------------- |
> >      | IT-att   [1]          |        $88.70 \pm 0.50$     |
> >      | TAM                |        $90.02 \pm 0.44$     |
> >
> >[1] Wang, et al. "Continual learning with lifelong vision transformer." In CVPR, 2022.
>
>
>
> #### Q3. Clarity of Tables 1 and 2 to incorporate columns displaying the number of parameters for each backbone model
>
> > - We appreciate this great suggestion, and in response, we have incorporated additional columns in Table 1 and 2 to present **the number of parameters (backbone + model-related)** for each method in the revision.
> > - We would like to humbly highlight that the observed improvements in performance are **not solely attributed to an augmented parameter count**, as supported by the following empirical evidence.
> >   - In the updated tables, we observe that FewTURE/HCTransformers, despite possessing a larger number of parameters, markedly lags behind the proposed MetaFormer.
> >   - We also conduct a comparative analysis with an ablated version, achieved by naively augmenting the number of layers in ViT-Small to make it comparable with the proposed MetaFormer. The results presented in the following table substantiate that merely increasing parameters cannot fully address the challenges inherent in few-shot learning. In fact, such augmentation may even elevate the risk of overfitting.
> >
> >
> > | Method                           | Backbone             | Backbone Params  | Total Params    | 1-shot           | 5-shot            |
> >| -------------------------------- | -------------------- | ---------------- | --------------- | ---------------- | ----------------- |
> >| vanilla ViT with more layers     | ViT-Small            | $21\mathrm{M}$   | $25.2 \mathrm{M}$ | $69.75 \pm 0.71$ | $84.12 \pm 0.56$  |
> >| MetaFormer-I (Ours)              | ViT-Small            | $21\mathrm{M}$   | $24.5 \mathrm{M}$ | $75.78 \pm 0.71$ | $90.02 \pm 0.44$  |
> >| MetaFormer-A (Ours)              | ViT-Small            | $21\mathrm{M}$   | $24.5 \mathrm{M}$ | $79.41 \pm 0.73$ | $91.21 \pm 0.44$  |
> >
> >[1] Hiller, et al. "Rethinking generalization in few-shot classification." In NeurIPS, 2022.
> >
> >[2] Lin, et al. "Supervised masked knowledge distillation for few-shot transformers." In CVPR, 2023

---

> ### Author Response · Authors · 2023-11-21
> **Response to Reviewer kzi2 (Part 3)**
>
> #### Q4. More in-depth discussions concerning prior research in the realm of meta-learning that incorporates vision transformers as their foundation architectures.
>
> > - Our claim that "existing meta-learning methods are specially tailored for CNNs and thus fail to translate effectively to vision transformers" is grounded in our empirical observations during the development of our proposed framework and supported by the findings in [5].
> >   - Initially, we had planned to **adapt FiLM**, a technique commonly employed in CNN-based meta-learning for task adaptation through conditioned batch normalization [2, 3], into **layer normalization layers of ViT for task conditioning**. Unfortunately, our experiments reveal **a performance drop** when ViT was applied with FiLM, as shown in the following table.
> >   - The work of **[5] also showcases the inferiority of FiLM when naively applied to ViT**, as compared to their proposed task conditioning method tailored specifically for ViT (which involves only conditioning the attention block with a bias), particularly on large-scale Meta-Dataset.
> >   - Such failures are attributed to the substantial differences between the two backbones [6].
> >
> > - We posit that the challenge of architectural inconsistency partially accounts for the **limited research in the realm of meta-learning grounded on ViT**. Another key challenge is the increased parameter requirement of ViT. FewTURE, as expounded in the Related Work section, is the pioneering work that tailors to ViT via inner-loop token importance reweighting, and addresses the second challenge via pre-training with DINO on the meta-training dataset. Our approach, empowering sample-to-sample and task-to-task interaction, further improves the accuracy substantially.
> > - To avoid misunderstanding, we have corrected the original claim to and incorporated these discussions into our supplemental material.
> >
> > | Method       | Backbone  | miniImageNet 5-way 1-shot |
> >| ------------ | --------- | ------------------------- |
> >| Vanilla ViT  | ViT-Small | $69.03 \pm 0.71$          |
> >| ViT + FiLM   | ViT-Small | $58.75 \pm 0.73$          |
> >| MetaFormer-I | ViT-Small | $75.78 \pm 0.71$          |
> >| MetaFormer-A | ViT-Small | $79.41 \pm 0.73$          |
> >
> > [1] Han-Jia, et al. "Few-shot learning via embedding adaptation with set-to-set functions." In CVPR, 2020.
> >
> > [2] Requeima, et al. “Fast and ﬂexible multi-task classiﬁcation using conditional neural adaptive processes.” In NeurIPS, 2019.
> >
> > [3] Oreshkin, et al. “TADAM: task dependent adaptive metric for improved few-shot learning.” In NeurIPS, 2018
> >
> > [4] Hiller, et al. "Rethinking generalization in few-shot classification." In NeurIPS, 2022.
> >
> > [5] Xu, et al. “Exploring Eﬃcient Few-shot Adaptation for Vision Transformers.” In TMLR, 2023
> >
> > [6] Raghu, Maithra, et al. "Do vision transformers see like convolutional neural networks?." In NeurIPS, 2021.

---

> ### Author Response · Authors · 2023-11-22
> **We would love to hear back from Reviewer kzi2**
>
> Hi Reviewer kzi2,
>
> We would like to follow up to see if our response addresses your concerns or if you have any further questions. We would really appreciate the opportunity to discuss this further if our response has not already addressed your concerns.
>
> Thank you again!

---

### Author Response · Authors · 2023-11-21
**Summary of changes**

We extend our sincere thanks to the reviewers for their constructive feedback. We have summarized additional experiments and clarification made during the rebuttal period as follows. The revised part has been highlighted in red color.


**Clarification:**
1. Illustrated our technical contributions concerning sample-level attention and task-level attention. (Reviewer kzi2 Q1 and Reviewer xYfp Q2)
2. Included the number of parameters for each backbone in Table 1 and 2 (Reviewer kzi2 Q3)
3. Corrected the typos in line 1 of page 5. (Reviewer 5apc Q3)
4. Added discussion of more related works. (Reviewer kzi2 Q2 and Q4)
5. Updated terminology used in the paper.  (Reviewer xXZf Q1)
6. Clarified the inductive and autoregressive setting. (Reviewer xXZf Q2)

**Additional Experiments:**
1. Conducted a comparative analysis between our proposed TAM method and the inter-task attention module, demonstrating TAM's superiority.  (Reviewer kzi2 Q2)
2. Analyzed the effectiveness of directly translating previous CNN-based meta-learning methods to vision transformers (Reviewer kzi2 Q4)
3. Performed ablation studies for hyper-parameter selection and more model variants to validate our contribution not solely due to an increase in parameters. (Reviewer 5apc Q2 and Reviewer 5apc Q1)
4. Applied our approach to CLIP, achieving remarkable success. (Reviewer xYfp Q1)
5. Provided a detailed computational analysis, including inference latency, GFLOPs and memory usage. (Reviewer 5apc Q2 and Reviewer xYfp Q3 and Q4)